# Exploiting a Y chromosome-linked Cas9 for sex selection and gene drive

Stephanie Gamez [1,9,11], Duverney Chaverra-Rodriguez [1,11], Anna Buchman [1,10], Nikolay P. Kandul[1], Stelia C. Mendez-Sanchez [1,2], Jared B. Bennett [3,4], Héctor M. Sánchez C. [4], Ting Yang [1], Igor Antoshechkin [5], Jonny E. Duque [1,6], Philippos A. Papathanos[7], John M. Marshall [4,8] & Omar S. Akbari [1✉]

CRISPR-based genetic engineering tools aimed to bias sex ratios, or drive effector genes into animal populations, often integrate the transgenes into autosomal chromosomes. However, in species with heterogametic sex chromsomes (e.g. XY, ZW), sex linkage of endonucleases could be beneficial to drive the expression in a sex-specific manner to produce genetic sexing systems, sex ratio distorters, or even sex-specific gene drives, for example. To explore this possibility, here we develop a transgenic line of *Drosophila melanogaster* expressing Cas9 from the Y chromosome. We functionally characterize the utility of this strain for both sex selection and gene drive finding it to be quite effective. To explore its utility for population control, we built mathematical models illustrating its dynamics as compared to other state-of-the-art systems designed for both population modification and suppression. Taken together, our results contribute to the development of current CRISPR genetic control tools and demonstrate the utility of using sex-linked Cas9 strains for genetic control of animals.

[1] Division of Biological Sciences, Section of Cell and Developmental Biology, University of California, San Diego, La Jolla, CA 92093, USA. [2] Group for Research in Biochemistry and Microbiology (Grupo de Investigación en Bioquímica Y Microbiología-GIBIM), School of Chemistry, Universidad Industrial de Santander, Bucaramanga, Colombia. [3] Biophysics Graduate Group, University of California, Berkeley, CA 94720, USA. [4] Divisions of Epidemiology & Biostatistics, School of Public Health, University of California, Berkeley, CA 94720, USA. [5] Division of Biology and Biological Engineering (BBE), California Institute of Technology, Pasadena, CA 91125, USA. [6] Centro de Investigaciones en Enfermedades Tropicales – CINTROP, Facultad de Salud, Escuela de Medicina, Departamento de Ciencias Básicas, Universidad Industrial de Santander, Piedecuesta, Santander, Colombia. [7] Department of Entomology, Robert H. Smith Faculty of Agriculture, Food and Environment, Hebrew University of Jerusalem, Rehovot 7610001, Israel. [8] Innovative Genomics Institute, University of California, Berkeley, CA 94720, USA. [9] Present address: Agragene Inc., San Diego, CA 92121, USA. [10] Present address: Verily Life Sciences, South San Francisco, CA 94080, USA. [11] These authors contributed equally: Stephanie Gamez, Duverney Chaverra-Rodriguez. ✉email: oakbari@ucsd.edu

The capacity to encode CRISPR-based gene-editing components within insect genomes, coupled with the ability to precisely orchestrate endogenous activity in every individual containing them, has inspired the exploration of novel methods that can modify insects in the population level[1]. For example, CRISPR-mediated genetic engineering can be used for large-scale manipulation of laboratory-reared individuals to produce phenotypes useful for genetic sexing, or male (♂) sterility, prior to release[2,3]. CRISPR-mediated modifications can also be used in sustainable suppression strategies aimed at reducing or eliminating wild populations, or in modification strategies directed to increasing the frequency of a desired allele or genotype, such as pathogen resistance, within a population. Among these, CRISPR-based gene drive strategies for either population suppression, or modification, are particularly promising and are presently under development by many groups[1,4–6].

In general, most of the control strategies directed to populations of insects that are vectors of human or plant diseases are focused on controlling females (♀'s), both because the number of ♀'s is determinant for population growth and because ♀'s are notoriously responsible for causing damage and spreading diseases. For these reasons, population suppression strategies that interfere with ♀ development, or ♀ fertility[7,8], or those that shift the sex ratio towards ♂'s[9–13], are among the most promising genetic control strategies being explored. Over the last decade, most of these efforts have focused on manipulating genetic elements located on the autosomes (chromosomes not involved directly in sex determination) as opposed to those located on the sex chromosomes[7,9,10]. Choosing autosomes over sex chromosomes makes practical sense, since autosomes typically contain more target genes and more highly-conserved regions than sex chromosomes, and are better-characterized and thus predictable. Sex chromosomes, on the other hand, are often gene-poor, repeat-rich, heterochromatic, silenced, unassembled, and more rapidly evolving than autosomes[14]. However, since sex bias is important for insect control and the rules governing sex chromosome inheritance are widely conserved (for example, an XY sex-determination system), linkage or targeting of CRISPR activity specifically to sex chromosomes can be advantageous for engineering specific types of genetic control mechanisms including gene drives[15–17].

One such proposed strategy involves the insertion of CRISPR elements on the ♂-specific Y chromosome to restrict activity exclusively to ♂'s, which can be beneficial, for example, for engineering gene drives with limited persistence or invasiveness[17], or for limiting undesired activity in the maternal germline that can adversely affect the spread of gene drives by generating functional resistant alleles in the female germline[2,18–21]. This can, in theory, be alternatively achieved by expressing these components from autosomes using ♂ germline-restricted regulatory sequences. However, in some cases, promoters that can ensure strict expression during early ♂ germline development have not been characterized sufficiently for use, perhaps because early gametogenesis is strikingly similar between the ♂ and ♀ germlines and involves shared genes and regulatory elements. Another appealing possibility of using sex chromosomes for insect control is the development of synthetic sex ratio distorters (SRDs)[13,15,22]. These include SRDs that have been successfully developed in a number of insect species using Cas9 expressed by sperm-specific promoters integrated on autosomes to target sequences on the X chromosome during spermatogenesis, resulting in biased transmission of X chromosome bearing gametes[10–12,23]. However, linking these proof-of-concept SRDs to the Y chromosome is necessary for making them truly applicable to the insect genetic control designs currently in development such as X-shredders or X-poisoning[12]. Although previously described SRDs[10–12,23] can be somewhat effective without being Y-linked, scaling and maintaining these SRD traits within a target insect population will require multiple releases of transgenic individuals, rendering this approach costly. On the other hand, SRDs which are Y-linked can reduce this cost considerably[24]. In the case of prezygotic elimination of X-bearing gametes, a strategy called CRISPR-based X-shredding could result in the first successful development of a Y chromosome meiotic drive, which is predicted to be one of the most rapid and resilient gene drive strategies described[10]. In the case of postzygotic elimination of daughters, a strategy now called CRISPR-based X-poisoning could result in a control method that remains non-invasive but is more persistent than other self-limiting strategies[17].

To begin exploring the potential of linking and/or targeting CRISPR activity to sex chromosomes for the first time in insects, we have recently established a method to insert transgenes on the *Drosophila melanogaster* Y chromosome using CRISPR-based homology-directed repair (HDR) insertion[25]. Here, we demonstrate that we can exploit this method to insert a transgene expressing the Cas9 endonuclease from the *vasa* regulatory regions into an intergenic region of the *D. melanogaster* Y chromosome. We explore the expression and efficiency of Cas9 expression from the Y and compare it to expression from other chromosomes, including the X chromosome. We demonstrate that it can induce robust endonuclease activity against endogenous target sequences, including essential and phenotypic genes. We evaluate its ability to function as a high-throughput reliable sex selection tool which we term S̲Ex̲ L̲ink̲E̲d C̲RISPR selec̲T̲ion (SELECT) and as a gene drive using a previously described system termed HomeR[26]. Finally, we use mathematical models to explore the utility of inserting Cas9 and other drive components on the Y chromosome, including applications to both population modification and suppression drive technologies. Taken together, this work paves the foundation for endonuclease expression on the Y chromosome of insects and enables further exploration of additional genetic control strategies that rely on Y-linked expression.

## Results

**Development of a Y chromosome-linked Cas9 strain.** Despite the availability of numerous transgenic fly lines in *D. melanogaster*, including multiple Cas9−expressing lines, there is a complete lack of transgenic lines expressing Cas9 protein from the Y chromosome. To engineer flies expressing Cas9 from the Y chromosome, we generated three plasmids containing SpCas9 driven by the *vasa* promoter which is expressed in both the ♂ and ♀ germline and soma[18]. As the Y chromosome is notoriously gene-poor and consists of silenced repetitive DNA[27], we incorporated a marker to assess promoter activity as done previously[18,28]. Downstream of the *vasa* promoter we included a SpCas9−T2A-eGFP cassette which encodes SpCas9 as well as a self-cleaving T2A peptide and an eGFP-coding sequence. We also incorporated a tdTomato transformation marker driven by the eye-specific 3xP3 promoter and flanked by *gypsy* and CTCF insulators to improve overall expression levels by acting as barrier elements that can block the propagation of heterochromatic structures into adjacent euchromatin[29]. These components were surrounded by homology arms to aid Y chromosome insertion into three distinct locations via CRISPR-mediated HDR (Only SGyA HDR is shown in Fig. 1a and b)[25]. To facilitate insertion, these transgenes were injected, along with an in vitro transcribed sgRNA complexed with recombinant SpCas9 protein into a fly line encoding a genetic source of *Nanos*-SpCas9[30]. Unfortunately, no transformants were obtained for SGyB and SGyC constructs (Fig. 1c). Of the 540 embryos injected for SGyA, 53% survived, and subsequent outcrossing to w[1118] (WT) resulted in a single

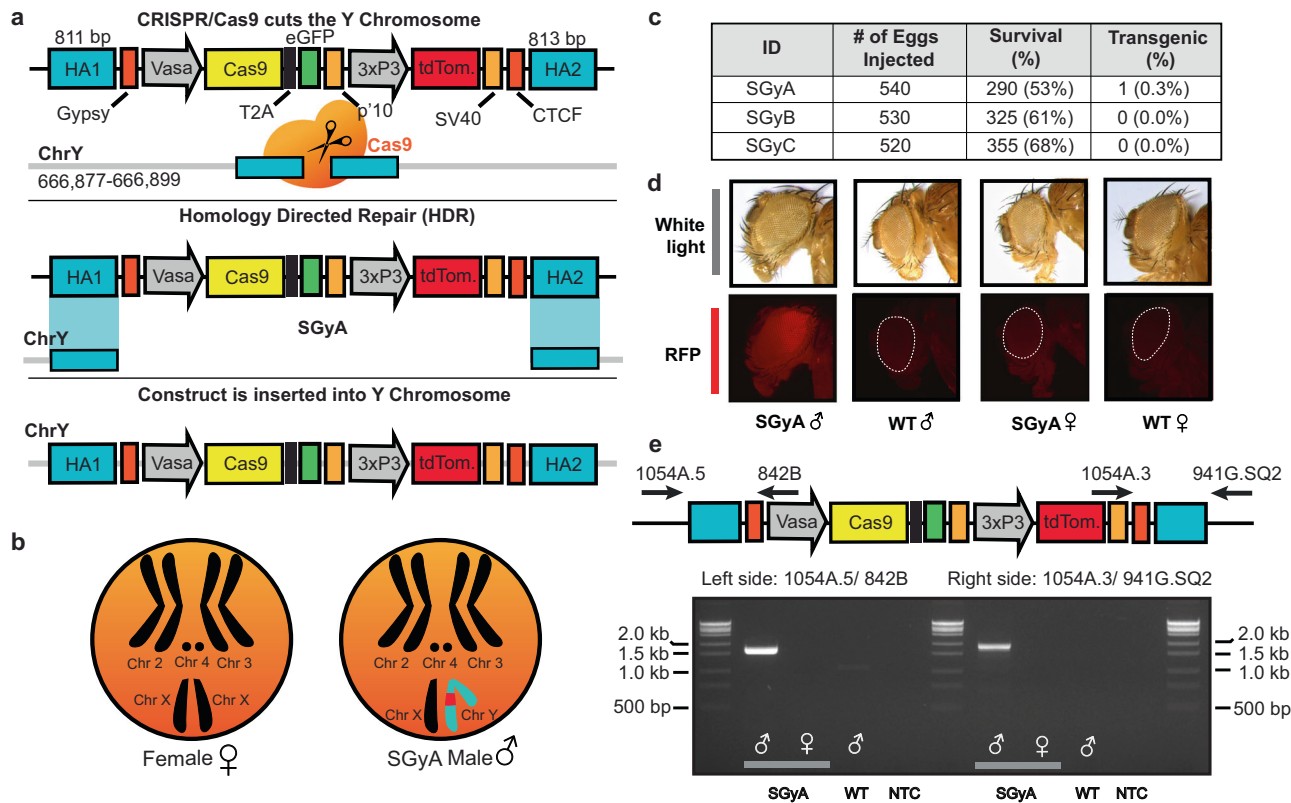

**Fig. 1 Engineering a Y chromosome-linked Cas9 in *D. melanogaster*. a** The Cas9 transgene design for Y chromosome insertion via CRISPR/Cas9—mediated cleavage and homology-directed repair (HDR). Homology arms (HA) flank two insulator sequences, GypSy and CTCF, and a vasa-controlled Cas9—T2A-eGFP. An eye-specific marker (Tdtomato) allows for the identification of transgenic flies carrying the transgene. For microinjection, a source of Cas9 and gRNA were provided to cleave the Y chromosome. The SGyA template was provided and inserted into the Y chromosome through HDR. **b** Karyotype of transgenic SGyA males and non-transgenic females. **c** The number of embryos injected for the SGyA, SGyB, and SGyC transgenes, survival to the larval stage of injected embryos, and rate (no. of independent transgenic individuals found/no. of embryos injected) is shown. No transformants were obtained for SGyB and SGyC. **d** White light and fluorescent images of the SGyA line compared to WT. There is an overall faint yet distinguishable expression of the fluorescent marker in SGyA individuals. Please refer to Fig. S1 to see the range of fluorescence in the SGyA eye marker. **e** PCR confirmation of transgenic male flies harboring the SGyA transgene. Primers corresponding to the left and right genomic insertion regions were used to amplify both sides to ensure the transgene was present. Amplification was performed at least twice by two independent scientists. The expected band size for the left side primer pair is 1.690 kb. The expected band size for the right side primer pair is 1.893 kb.

transformant G1 ♂ (Fig. 1c, d). To expand this line, and remove the genetic source of *Nanos*-spCas9, this G1 ♂ was outcrossed to WT ♀'s. In agreement with the paternal inheritance of the Y chromosome, we observed 100% of the G2 transgenic ♂'s, with no ♀'s, expressing the tdTomato eye marker (Fig. 1d). A stock containing the Y chromosome-linked Cas9 transgene was established, and from hereon is referred to as SGyA, displaying exclusive ♂-specific inheritance patterns which continued into all subsequent generations. We molecularly verified the presence of the transgene by PCR and Sanger sequencing across the transgene-insertion junctions using genomic DNA of SGyA ♂'s (Fig. 1e). In establishing the SGyA stock, we observed a range of variable eye maker expressions in adult ♂'s that ranged from moderate to undetectable fluorescence, presumably resulting from heterochromatic silencing effects (Supplementary Figure 1A). Despite this range in marker expression, weak and moderate marker expressing males still displayed sufficient Cas9 activity that was capable of editing and generating high rates of lethality in males that inherited a sgRNA transgene targeting *PolG2* (Supplementary Figure 1C). We, therefore, maintained the SGyA line by allowing the ♂s with varied fluorescent marker expression to mate with WT ♀s each generation (Supplementary Figure 1B).

Given that the SGyA line encodes a T2A-eGFP marker, similar to an available autosomal linked *Vasa*-Cas9—T2A-eGFP line[2], we were able to visually compare expression levels between these two strains. To do this, 3–4 day old ♂ testes (structure depicted in Supplementary Figure 2A) were dissected and imaged to assess relative eGFP expression levels comparing WT (negative control) (Supplementary Figure 2B, B'), autosomal linked Cas9 (Supplementary Figure 2C, C'), and SGyA (Supplementary Figure 2D, D'). We expected eGFP expression to manifest in the testes (long structures curled around seminal vesicles; Supplementary Figure 2) due to the role of *vasa* in germline development[31]. As expected, no eGFP was detected in WT testes (Supplementary Figure 2B'), but visible eGFP fluorescence was present in both autosomal *Vasa*-Cas9 and SGyA testes and seminal vesicles indicating robust expression of the transgenes in the ♂ germline (Supplementary Figure 2C' and Supplementary Figure 2D', respectively).

**Quantification of Cas9 expression.** To quantify the expression of the Y-linked *vasa*-Cas9, we performed RNA-sequencing from 3–4 day old ♂'s, using WT ♂'s and autosomal linked *vasa*-Cas9 as negative and positive controls, respectively. We detected robust expression of the dsRed and eGFP markers and Cas9 in

autosomal samples, with comparatively lower expression of tdTomato, eGFP, and Cas9 from SGyA samples, with no significant expression observed in control WT samples (Supplementary Data File 1). A DeSeq analysis revealed that autosomal Cas9 transgenic flies have 5858 differentially expressed genes (Supplementary Figure 3A) and SGyA transgenic flies have 476 differentially expressed genes (Supplementary Figure 3B, Supplementary Data File 2 and Supplementary Data File 3, respectively). Of the list of differentially expressed genes in both samples, 321 genes were found to be the same between samples. To get a sense of what genes are differentially expressed, we also included a Gene Ontology analysis (GO) in our DeSeq output. The top upregulated genes in the autosomal Cas9 vs WT dataset (log2FoldChange) include the DsRed and eGFP markers, Cas9, several long non-coding RNAs, genes associated with defense response, cuticle development, proteolysis, and Hsp70 (Heat-shock proteins) among others (Supplementary Data File 2). In the SGyA vs WT dataset, top upregulated genes include Cas9, GFP, and tdTomato markers, several long non-coding RNAs, defense response genes, and proteolysis genes (Supplementary Data File 3). Taken together, these data confirmed expression from the Y-linked *vasa*-Cas9, albeit it was slightly weaker than the autosomal linked *vasa*-Cas9.

**Gene editing using the Y-linked *vasa*-Cas9.** To genetically assess the efficacy of the SGyA line, we crossed SGyA ♂'s to ♀'s from strains encoding gRNAs targeting genes that result in clear visual phenotypes when disrupted. For example, we used an available strain simultaneously expressing multiplexed sgRNAs targeting four genes including sepia, ebony, curled, and forked, each flanked by tRNA's (tRNA-sgRNA) from a single promoter[32] (Fig. 2a, b). We also tested five additional strains encoding sgRNAs targeting wingless, cut, apterous, twisted, and scalloped (Supplementary Figure 5)[33]. To compare the effects of the Y chromosome linkage on Cas9 activity we used the autosomal *vasa*-Cas9 as a positive control[18]. In crosses with a genetic source of Cas9, mutant phenotypes were seen in the F1 generation, whereas no phenotypes were observed in WT crosses lacking Cas9 (Fig. 2c and Supplementary Figure 4A, Supplementary Figure 5). In experimental crosses involving SGyA ♂'s and the tRNA-sgRNA, we found that ♀ F1 progeny did not inherit the Cas9 transgene, and therefore did not display mutant phenotypes as expected. F1 ♂'s exhibited subtle mosaic mutant phenotypes in three out of the four target genes, showing that activity of the Y-linked Cas9 is specific to ♂'s (Fig. 2c and Supplementary Figure 4B; Table 1). No mutants were recovered for the *cu* target and were therefore excluded from the figure. A greater proportion of the mutant ♂'s were single mutants for the forked gene. Typically, forked null mutants tend to have several bristles that are short and have split ends. In the forked mutants, we saw a few bristles that were short, and even fewer bristles had split ends. In the case of ebony, null mutants have a dark cuticle, however, in our mutant ♂'s, we observed mosaic patches of ebony cuticle on the thorax (Supplementary Figure 4B). The inheritance of the SGyA transgene was PCR confirmed in F1 ♂'s for all crosses (Supplementary Figure 6A). In comparison, an autosomal source of Cas9 produced F1 ♂'s and ♀'s with increased penetrance and expressivity of mutant phenotypes; typically producing triple mutants (ebony-forked-sepia mutants ~92%). (Fig. 2c and Supplementary Figure 4C; Table 1). For example, F1 progeny from autosomal crosses had several mosaic patches of the dark cuticle as compared to SGyA mutants. In addition, mutant phenotypes were seen for sepia, whereas none were seen for SGyA progeny. Interestingly, F1 progeny maintained a wild-type allele in the curled gene regardless of the source of Cas9, suggesting there was

reduced cleavage at this target site. The sequences derived from the F1 progeny in the autosomal crosses revealed indels in the genes ebony, and sepia, (Supplementary Figure 7). For the progeny derived from SGyA crosses, sequenced target sites showed the wild-type alleles with multiple peaks suggesting somatic mosaicism in the individuals. Further subcloning of the PCR amplicons and sequence analysis showed separated indels for ebony but not curled, forked, or sepia. Crosses between homozygous sgRNA ♀'s and WT ♂'s showed no mutant phenotypes as expected (Supplementary Figure 5B–F, 5B'–F').

In crosses with ♂s with an autosomal source of Cas9 and ♀ with the single U6:sgRNAs, both F1 ♂ and ♀ progeny displayed phenotypes. For example, when twisted is targeted, both ♂'s and ♀'s have twisted abdomens (Supplementary Figure 5B"). We observed embryo/early larvae and pupae lethality when cut and wingless were targeted (Fig. 5c" and S5D", respectively). When apterous (*ap*) and scalloped (*sd*) were targeted, both ♂'s and ♀'s were affected (Supplementary Figure S5E, F; Supplementary Figure 5E", S5F"). However, in *ap*, a small proportion of ♂'s and ♀'s retained WT phenotypes (Supplementary Figure 5E"). In experimental crosses involving SGyA ♂'s and the single U6:sgRNA expression system, we found that ♀ F1 progeny did not display mutant phenotypes as expected. Sequencing the target sites from these females showed no indels (Supplementary Figure 8). Crosses between SGyA ♂'s and sgRNA ♀'s targeting twisted (*tw*) produced ♂'s with disfigured abdomens which affected the numbers of F1 ♂'s emerging (Supplementary Figure 5B, 5B") compared with crosses with WT ♂'s lacking a source of Cas9. Crosses from the SGyA and lines expressing gRNAs for cut (*ct*) and wingless (*wg*) caused ♂ lethality at the embryo/early larvae and pupae, respectively, and produced exclusively adult F1 ♀'s (Supplementary Figure 5C, D; Supplementary Figure 5C", 5D"). Crosses with lines expressing gRNAs targeting ap and sd produced wing deformities that prevented ♂'s from emerging from the puparium, affecting the final number of adults counted (Supplementary Figure 5E, F; Supplementary Figure 5E", F"). Taken together, these data suggest our SGyA line is able to efficiently produce mutant phenotypes in a ♂ specific manner.

**Efficient sex selection by exploiting sex chromosome-linked Cas9.** Given the efficiency of Lethal Biallelic Mosaicism (LBM)[2,3,18,26], we hypothesized that this mechanism could be exploited, in combination with sex chromosome-linked Cas9 elements, as a novel method for sex selection that we term SEx LinkEd CRISPR selecTion (SELECT). To explore this hypothesis, we opted to target an essential haplosufficient gene, DNA Polymerase gamma subunit 2 (*PolG2*, DNA polymerase γ 35 kDa, CG33650) required for the replication and repair of mitochondrial DNA[34]. Importantly, high levels of biallelic somatic mosaicism of *PolG2* are lethal[26]. To compare cleavage efficiencies, we outcrossed either WT, SGyA, autosomal Cas9, or X-linked Cas9 ♂'s to a 2nd chromosome-linked strain expressing a sgRNA targeting *PolG2* (referred to as U6.3-gRNA#1[PolG2]) driven by the U6.3 promoter[2].

When WT ♂'s are outcrossed to heterozygous U6.3-gRNA#1-[PolG2] ♀'s, four expected F1 phenotypes are observed; meanwhile, homozygous U6.3-gRNA#1[PolG2] ♀'s produce only two expected phenotypes and the surviving F1 individuals correspond with the genotypes expected from these crosses (Fig. 3a, b; Table 2). When autosomal *vasa*-Cas9 ♂'s are crossed with heterozygous U6.3-gRNA#1[PolG2] ♀'s, all transheterozygous F1 progeny perish, while F1 ♀'s (w−;CyO+;Cas9+) and ♂'s (w+;CyO+;Cas9+) expressing Cas9 but lacking the gRNA transgene were recovered (Fig. 3c; Table 2). The crosses involving homozygous U6.3-gRNA#1[PolG2]

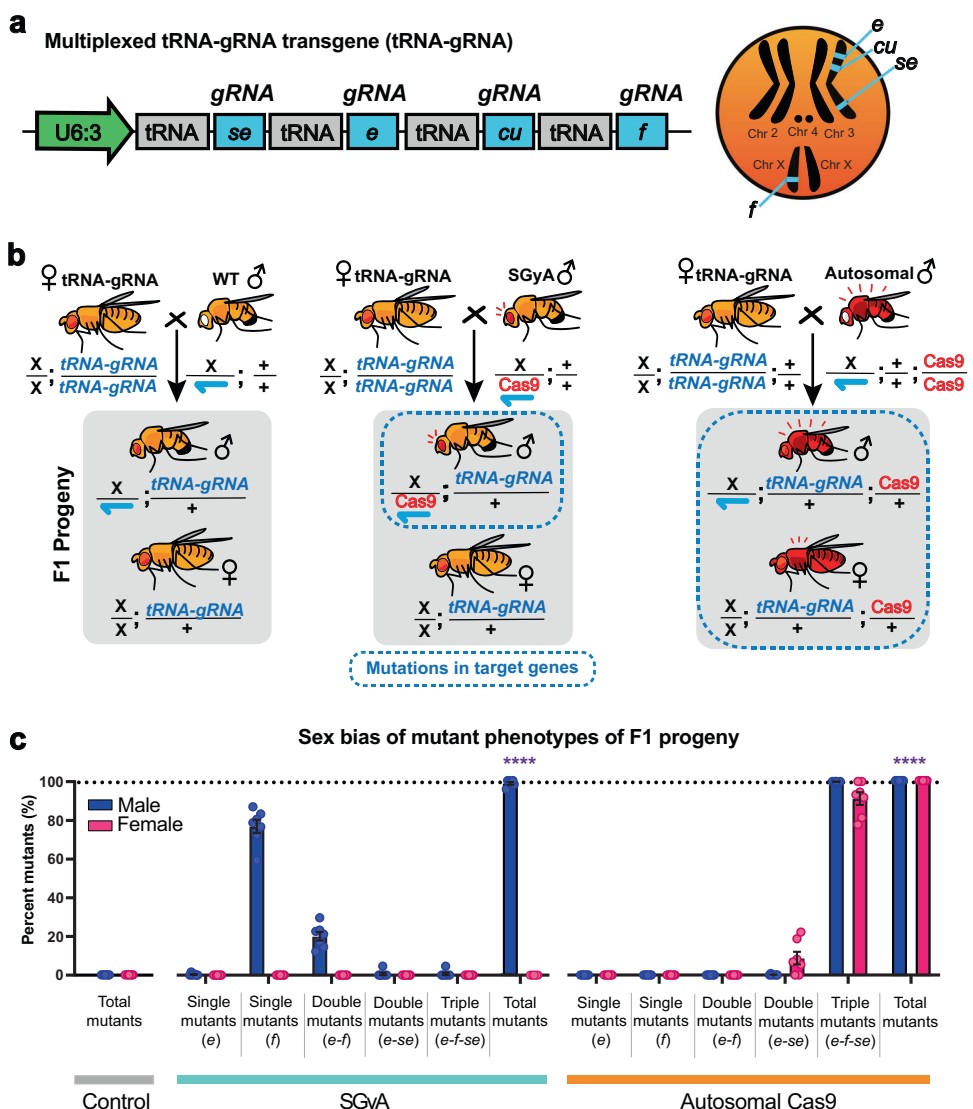

**Fig. 2 Functionality of the SGyA with a multiplexed tRNA-gRNA system. a** The multiplexed tRNA-gRNA transgene was used to determine the functional capacity of a Y-linked Cas9 to cleave four phenotypic genes, sepia (se), ebony (e), curled (cu), and forked (f). Flanking the gRNA's with tRNAs enables expression from a single promoter and processing of the multiplexed gRNAs. Karyotype on the right depicts the location of the four target sites. No cu mutants were obtained and were thus omitted. **b** Crossing schematic of experiment. Homozygous tRNA-gRNA females were outcrossed to either WT, SGyA, or autosomal Cas9 males. F1 progeny were expected to have either no phenotype and/or a range of mutant phenotypes. Those expected to have a phenotype are indicated in blue dashed borders. **c** Percentages of F1 progenies with single mutations, double mutations, or triple mutations. F1 progeny from genetic crosses involving SGyA demonstrated single, double, and triple mutants with only 38% of the progeny being composed of f mutants. Crosses involving an autosomal source of Cas9 produced mainly triple mutants. For experimental crosses, seven replicate crosses were set up. For the control, only five replicate crosses were set up. A two-way ANOVA with Tukey's multiple comparisons was performed on the total mutants (both male and female) comparing wild type and SGyA or autosomal Cas9 data to determine significance. Error bars in black represent the mean ±SEM. ****$p < 0.0001$. Source data is provided as a Source Data file.

♀'s, and ♂'s as a source of autosomal Cas9 produced 100% lethality of all F1 regardless of sex, since all the progeny inherited the U6.3-gRNA#1[PolG2] from the ♀'s and the Cas9 from the ♂'s and were subjected to LBM (Fig. 3d; Table 2). We next crossed ♂'s from X-linked Nanos Cas9 (Bloomington Fly Stock line # 54591) to heterozygous U6.3-gRNA#1[PolG2] ♀'s, and all transheterozygous ♀'s died. From this cross, we recovered only F1 ♂'s lacking Cas9 (w-;CyO+;Cas9− and w+;U6.3-gRNA#1[PolG2]/CyO−;Cas9−); and F1 ♀'s (w+;CyO+;Cas9+) expressing Cas9 but lacking the gRNA transgene (Fig. 3e). The crosses involving homozygous U6.3-gRNA#1[PolG2] ♀'s produced 100% ♂'s since all F1 ♀'s inherited the X chromosome-Cas9 from the ♂'s and the

gRNA from the U6.3-gRNA#1[PolG2] ♀'s (Fig. 3f; Table 3). Finally, when SGyA ♂'s are crossed with heterozygous U6.3-gRNA#1-PolG2 ♀'s, only ♀'s (w-;CyO+;Cas9− and w+;U6.3-gRNA#1[PolG2]/CyO−;Cas9−), and F1 ♂'s (w+;CyO+;Cas9+) lacking the gRNA transgene were recovered, since all transheterozygous F1 ♂'s died (Fig. 3g; Table 3). Similarly, the crosses involving homozygous U6.3-gRNA#1[PolG2] ♀'s with SGyA ♂'s produced ~98.6% viable ♀'s (w+;U6.3-gRNA#1[PolG2]/CyO−;Cas9−) and 1.4% ♂'s (Fig. 3h; Table 3). Taken together, these results demonstrate a novel and efficient genetic sexing technique by exploiting LBM (Fig. 3i) using sex-linked Cas9 lines crossed to homozygous gRNA lines targeting essential genes.

**Table 1 ANOVA comparisons of cleavage activity among experimental groups.**

| Strain | | | |
|---|---|---|---|
| **Phenotype of F1 progeny** | **tRNA-gRNA (N)** | **SGyA (N)** | **Autosomal Cas9 (N)** | **P value** |
| WT phenotype | 100 ± 0% (5; 505) | 50.5 ± 1.2% (6; 1218) | 0 ± 0% (5; 907) | <0.0001 |
| Ebony | 0 ± 0% (5; 505) | 0.2 ± 0.1% (6; 1218)[*] | 0 ± 0% (5; 907) | 0.1512 |
| Forked | 0 ± 0% (5; 505) | 38.6 ± 1.9% (6; 1218)[*] | 0 ± 0% (5; 907) | <0.0001 |
| Ebony-Forked | 0 ± 0% (5; 505) | 10.1 ± 1.2% (6; 1218)[*] | 0 ± 0% (5; 907) | <0.0001 |
| Ebony-Sepia | 0 ± 0% (5; 505) | 0.3 ± 0.3% (6; 1218)[*] | 4.9 ± 2.1% (5; 907) | 0.0156 |
| Ebony-Forked-Sepia | 0 ± 0% (5; 505) | 0.3 ± 0.3% (6; 1218)[*] | 95.1 ± 2.1 (5; 907) | <0.0001 |

[*]Recovered only males.
Distinct differences between the activity of a Y-linked Cas9 and an autosomal Cas9 can be observed in the percentages of individuals with particular phenotypes. In SGyA experimental crosses, only males (with an asterisk) had mutant phenotypes. females were unaffected. The majority of F1 progeny from autosomal crosses had triple mutant phenotypes. Average percentages and SEM values are listed below. The number of replicates is listed next to the total number of F1 progeny counts. Significance was determined using a two-tailed student's t test.

**Characterization of the Y-encoded *vasa*-Cas9 fly line as a split gene drive**. To validate the utility of SGyA in a split gene drive context (Table 3), we genetically crossed the SGyA transgenic flies with a previously generated HomeR gene drive element (GDe)[26]. The GDe was composed of a re-coded polymerase gamma sub-unit 2 rescue (*PolG2*, DNA polymerase Ɣ 35-kDa, CG33650), a *PolG2* gRNA, and a marker (3xp3-GFP) that enables scoring GDe inheritance. This transgene should permit the survival of flies inheriting the GDe and the lethality of flies that harbor biallelic NHEJ events. To assess its functionality as part of a sex-biased split drive, we performed genetic crosses (in pentaplicate), by crossing SGyA ♂'s to GDe ♀'s (Fig. 4a). Similar crosses with an autosomal source of Cas9 were also performed for comparison (Fig. 4b). Negative control was performed using WT ♂'s outcrossed to GDe ♀'s (Fig. 4c). Before assessing the inheritance rate of the GDe, we determined the hatching rate of embryos produced from crosses between GDe and WT, GDe and SGyA, and GDe and autosomal Cas9. We found no significant differences between the hatching rate of WT and SGyA (unpaired t test, p value = 0.1836), however, autosomal Cas9 crosses often produced fewer larvae compared to the control (unpaired t test, p value = 0.0094) (Fig. 4d).

In assessing the ability of the SGyA element to promote the non-Mendelian transmission of the GDe element, we observed that when F1 SGyA/GDe transheterozygous ♂'s were outcrossed to WT ♀'s, 65.3% of the F2 offspring, on average, inherited the GDe element (marked by the dominant GFP marker) compared to 50.6% of the F2 progeny in the negative control (F1 heterozygous GDe ♂'s outcrossed to WT ♀'s; p value = 0.0042, unpaired t test) (Fig. 4e). Similarly, when transheterozygous ♂'s (with an autosomal source of *vasa*-Cas9 and GDe) were outcrossed to WT ♀'s to assess GDe inheritance, we found that 69% of the F2 offspring on average inherited the GDe element. (Fig. 4e). This result is similar to the previous study which found that 63% of the offspring inherited the GDe in crosses involving a WT mother and a transheterozygous father containing an autosomal source of vasa-Cas9 and the GDe[26]. There were no significant differences in inheritance rates of the GDe transgene between SGyA and autosomal Cas9 experiments (p value = 0.337, unpaired t test) (Fig. 4e). A subsequent outcross was carried out with the F2 transheterozygous progeny to determine if the homing rate of the GDe changed in subsequent generations (in the F3 progeny). We did not observe a significant deviation of GDe inheritance frequencies between the F2 and F3 progeny data of SGyA and autosomal Cas9 (Respectively, Fig. 4f, g). Taken together, these data suggest SGyA can function as a split gene drive and has comparable drive efficiency in the male germline to an autosomal source of Cas9.

**Modeling indicates SGyA-based drive systems enact enduring population modification and rapid suppression**. Advancing upon the characterization of Y-encoded Cas9 functioning as a split gene drive, and the goal of utilizing Y-encoded Cas9 as a population suppression system, we performed modeling to explore the potential for SGyA-based drive systems to enact efficient population modification and suppression. We conducted population simulations using the MGDrivE framework[35], comparing the performance of the Y-linked systems to equivalent X-linked and autosomal systems (Fig. 5). We performed simulations for *Anopheles gambiae*, a mosquito disease vector that proof-of-concept gene-editing tools from *D. melanogaster* are often applied to[2,3]. Two populations with an equilibrium size of 10,000 were simulated, exchanging migrants at a rate of 1% per mosquito per generation[36]. For all drive systems, 12 consecutive weekly releases of 10,000 ♂ mosquitoes homozygous or hemizygous for each drive allele were simulated in the release population, and spread in both the release and neighboring populations was recorded.

We first compared the performance of an SGyA-based split drive, in which the Cas9 is Y-linked and the gRNA locus is autosomal, to split drive systems in which the Cas9 is: (i) at an unlinked autosomal locus, and (ii) X-linked (Fig. 5a). For standard *An. gambiae* life-history parameters and gene drive parameters from a split drive system engineered in another mosquito vector, *Ae. aegypti*[37] (confinable split drive systems have yet to be demonstrated in *An. gambiae*) (Supplementary Data File 4), modeling results suggest that 12 weekly releases are sufficient to drive the gRNA/effector allele (red) to high frequency in the release population (>95% of ♀'s having at least one copy of the effector gene) for all three split drive designs. The Cas9 allele (blue) then falls out of the population due to a fitness cost, and the gRNA/effector allele is slowly eliminated as it also has a fitness cost and its inheritance bias is dependent upon co-occurrence of the Cas9 and gRNA alleles. Two interesting distinctions between the Y-linked and autosomal/X-linked split drive systems are that: (i) the gRNA/effector allele persists in the population for longer for the Y-linked system, and (ii) the Y-linked system spreads to a higher frequency in the neighboring population. An important metric for population modification strategies is the "window of protection" (WoP), which we define here as the duration that ♀ mosquitoes having the anti-pathogen effector gene remain at a frequency of 90% or higher in the population. When Cas9 is Y-linked, we calculate a WoP of 672 days from our simulations, which is significantly higher than the WoP for the autosomal split drive (428 days) and when Cas9 is X-linked (344 days). Migrants having the gRNA/effector allele accumulate in the neighboring population. Consequently, for the Y-linked design, the gRNA/effector allele spreads to a higher frequency and persists for a

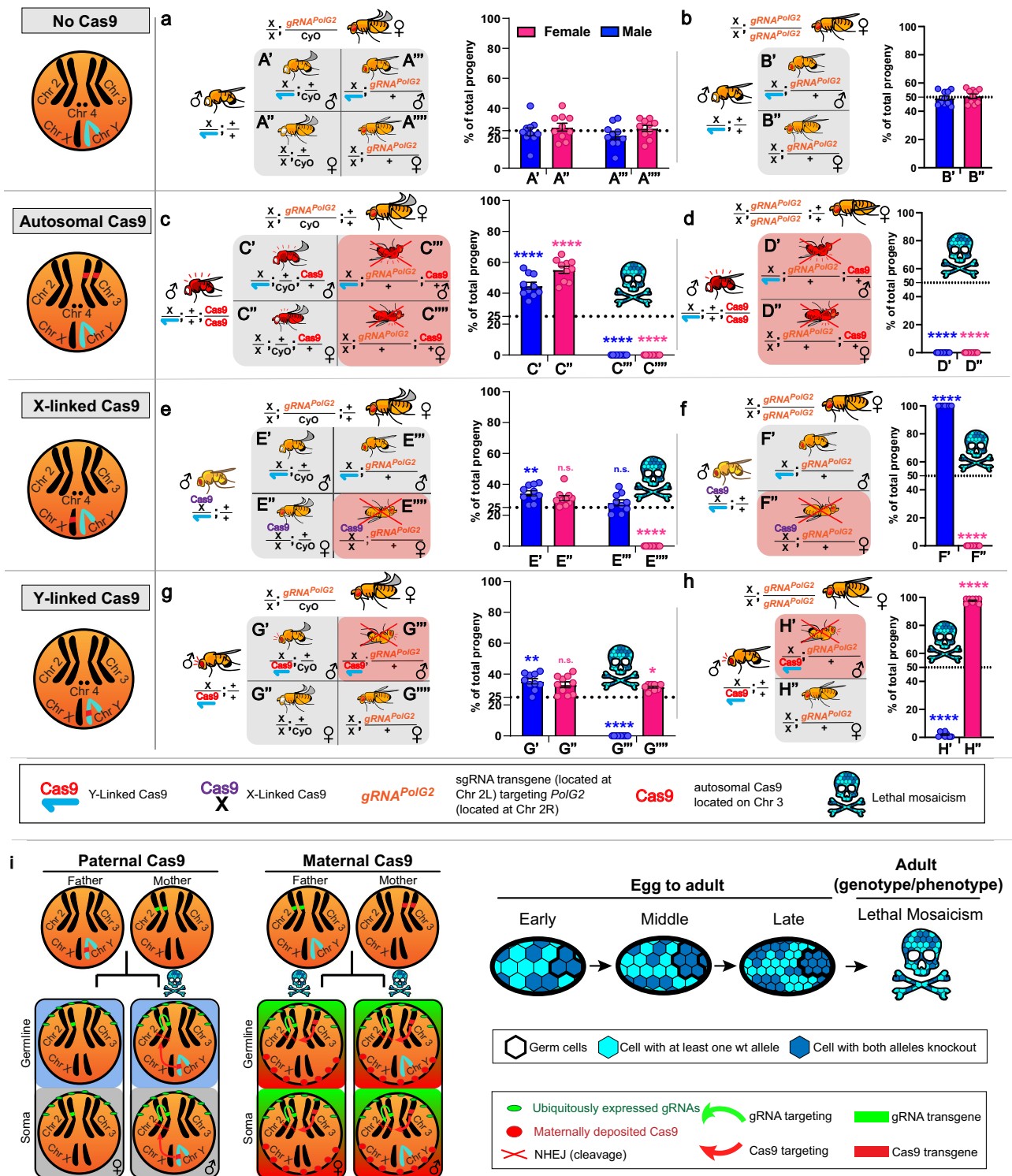

longer duration in the neighboring population, reaching a maximum carrier frequency (frequency of ♀ mosquitoes having at least one copy of the allele) of 48%, compared to 34% for the autosomal design and 25% for the X-linked design. The Y-linked design is, therefore, less confineable, although all three designs are self-limiting, meaning that spread in both the release and neighboring populations is transient.

Next, we compared the performance of an SGyA-based X-shredder, in which all drive components are Y-linked, to two other population suppression drive systems: (i) an autosomal X-shredder, and (ii) an autosomal homing-based drive targeting a

gene required for ♀ fertility[8] (Fig. 5b). The Y-linked X-shredder is a promising population suppression system for *An. gambiae* mosquitoes (for which ♂ are XY) as cutting of X gametes in the ♂ germline leads to an increasingly ♂ sex bias and potentially a population crash or persistent population suppression[15,38]. As this system spreads from a low population frequency, it could be effective over a wide geographic range. A general weakness of population suppression strategies is that drive-resistant alleles have a significant selective advantage, and hence if they emerge, the population is likely to rebound[39]. Assuming high rates of DNA cleavage (0.99 per heterozygote), as seen for other

**Fig. 3 Functionality of SGyA as a genetic sex-sorter termed SELECT using a single gRNA targeting PolG2.** In **a**, **c**, **e**, **g** Heterozygous U6.3-gRNA#1[PolG2] females were crossed to a male that was either WT, SGyA, or autosomal Cas9. Below the cross is the outcome of the F1 progeny. Each F1 fly with a corresponding genotype is associated with either X′, X″, X‴, or X″″ labels. The gRNA transgene is marked with a non-fluorescent orange-red eye maker. **a** No lethal phenotypes were seen in negative control crosses. **c** In autosomal Cas9 and gRNA crosses, both F1 transheterozygous males and females were not recovered due to the lethal effects of PolG2 cleavage. **e** In X-linked Cas9 crosses, both F1 transheterozygous males and females were not recovered. Only individuals without gRNA survived. **g** In SGyA crosses, only F1 transheterozygous males inheriting both SGyA and the U6.3-gRNA#1[PolG2] transgene resulted in lethality (as indicated with red "X"). Females survived. **b**, **d**, **f**, **h** Homozygous U6.3-gRNA#1[PolG2] females were outcrossed to a male that was either WT, SGyA, or autosomal Cas9. **b** No lethal phenotypes were seen in negative control crosses. **d** No surviving offspring were recovered in autosomal Cas9 crosses. **f** All F1 U6.3-gRNA#1[PolG2] males survived and F1 females perished in X-linked-Cas9 crosses. **h** All F1 transheterozygous males resulted in lethality while F1 females survived in SGyA outcrosses. **i** Mechanism and schematic depicting lethal mosaicism in progeny from paternal vs maternal inheritance of Cas9. Lethality is only observed in males which inherit Y-linked Cas9. However, all progeny perish when inheriting maternal Cas9 due to maternally deposited Cas9. Red shaded boxes in Punnett square represent lethality. Mosaic skulls represent lethal mosaicism. Instances, where this symbol is seen, represents no progeny was recovered. Blue and pink bars represent the percentage of males and females, respectively. Black bars represent the standard error of the mean(SEM). Ten replicate crosses were set up per experiment. A two-tailed unpaired student's $t$ test is used to determine the significance of percentages compared with WT. (refer to Table 2 for ANOVA comparisons) ****$p < 0.0001$; **$p < 0.005$; *$p < 0.05$. N.s. no significance. Source data is provided as a Source Data file.

**Table 2 Sex-specific lethality driven by Y-linked Cas9 and X-linked Cas9 disrupting the PolG2 gene using heterozygous females.**

| ♂ (Cas9) line crossed to ♀ gRNA[PolG2#1] w+/CyO− line (N = 10 crosses/line) | F1 progeny | | | | | | | |
|---|---|---|---|---|---|---|---|---|
| | Total | x̄ ± STDV | ♀w−/gRNA− | ♀w+/gRNA+ | ♂w−/gRNA− | ♂w+/gRNA+ | $\chi^2$ (3 DF) | P value |
| WT | 407 | 40.7 ± 19.3 | 102 | 103 | 103 | 99 | 0.1 | >0.95 |
| SGyA | 472 | 47.2 ± 15.6 | 154 | 148 | 170 | 0 | 159.5 | <0.001 |
| X-Cas9 | 783 | 78.3 ± 13.6 | 0 | 267 | 273 | 243 | 263.6 | <0.001 |
| Autosomal Cas9 | 425 | 42.5 ± 15.5 | 232 | 0 | 193 | 0 | 432.2 | <0.001 |

Differences in F1 survival are produced by the activity of the Y-linked Cas9 and autosomal Cas9 in transheterozygous individuals. Negative control used w[1118] males crossed to w+/CyO− females (heterozygous for gRNA#1[PolG2]). The number of individuals observed for each of the four phenotypes from this cross is expected by random assortment ($\chi^2 = 0.1$, $p > 0.95$). In SGyA experimental crosses, only transheterozygote males (w+/CyO+) were killed, while females (regardless of their phenotype) and males that inherited the Cas9 transgene but not the gRNA element (w−/CyO−) were not affected. Conversely, when females were crossed to males carrying an X-linked Cas9 transgene, only transheterozygous females were killed. The F1 progeny from autosomal Cas9−derived crosses produced only males and females who did not inherit the gRNA element (w−/CyO−), whereas all transheterozygote males and females (w−/CyO−) were differentially killed. Significance was determined using a one-tailed $\chi^2$ test.

**Table 3 Sex-specific lethality driven by Y-linked Cas9 and X-linked Cas9 disrupting the PolG2 using homozygous females.**

| ♂ (Cas9) line crossed to ♀ gRNA[PolG2#1] w+/gRNA[PolG2#1] w+ line (N = 10 crosses/line) | F1 progeny | | | | | |
|---|---|---|---|---|---|---|
| | Total | x̄ ± STDV | ♀w+/gRNA+ | ♂w+/gRNA+ | $\chi^2$ (1 DF) | P value |
| WT | 496 | 49.6 ± 16.7 | 249 | 247 | 0.01 | >0.95 |
| SGyA | 331 | 33.1 ± 19.9 | 326 | 5 | 311.3 | <0.001 |
| X-Cas9 | 200 | 28.6 ± 15.0 | 0 | 200 | 200 | <0.001 |
| Autosomal Cas9 | 0 | 0 | 0 | 0 | 0 | <0.001 |

Differences in F1 survival are produced by the activity of the Y-linked Cas9 and autosomal Cas9 in transheterozygous individuals. Negative control used w[1118] males crossed to w+/CyO− females (homozygous for gRNA#1[PolG2]). The number of individuals observed for each of the four phenotypes from this cross is expected by random assortment ($\chi^2 = 0.1$, $p > 0.95$). In SGyA experimental crosses, only transheterozygote males (w+/CyO+) were killed, whereas females (regardless of their phenotype) and males that inherited the Cas9 transgene but not the gRNA element (w−/CyO−) were not affected. Conversely, when females were crossed to males carrying an X-linked Cas9 transgene, only transheterozygous females were killed. The F1 progeny from autosomal crosses produced only males and females which did not inherit the gRNA element (w−/CyO−), whereas all transheterozygote males and females (w−/CyO−) were differentially killed. Significance was determined using a one-tailed $\chi^2$ test.

CRISPR-based drive systems in An. gambiae[7], and a low rate of resistant allele generation ($10^{-6}$ per heterozygote), as required for effective population suppression[40] (Supplementary Data File 4), we find that 12 weekly releases lead to population elimination in 99% of the simulations, which is reached on average 21 weeks after the final release. A notable difference for the autosomal homing-based drive targeting a ♀ fertility gene is that, for equivalent parameters (Supplementary Data File 4), it achieves population elimination with similar frequency (98% of simulations), but reaches elimination more slowly, on average 36 weeks after the final release. That said; for lower resistant allele generation rates ($<10^{-6}$ per heterozygote), both systems are expected to achieve population elimination ~100% of the time. Additionally, the Y-linked X-shredder, by targeting multiple genetic loci on the X chromosome simultaneously[10],

may be easier to limit resistant allele generation for. Lastly, the autosomal X-shredder is a self-limiting population suppression system that could be used as an alternative to SIT strategies, or as an intermediate technology prior to the release of a non-localized Y-linked X-shredder. Here, we see that 12 weekly releases of this autosomal system lead to transient population suppression at the release site (the population rebounds to 95% of its original size within 51 weeks) and only limited spread to the neighboring population.

## Discussion

The generation of Y-linked fluorescent marker lines has been reported in mosquitoes, flies, and mice[25,41,42]. Here, we report the first generation of a transgenic line that robustly expresses the

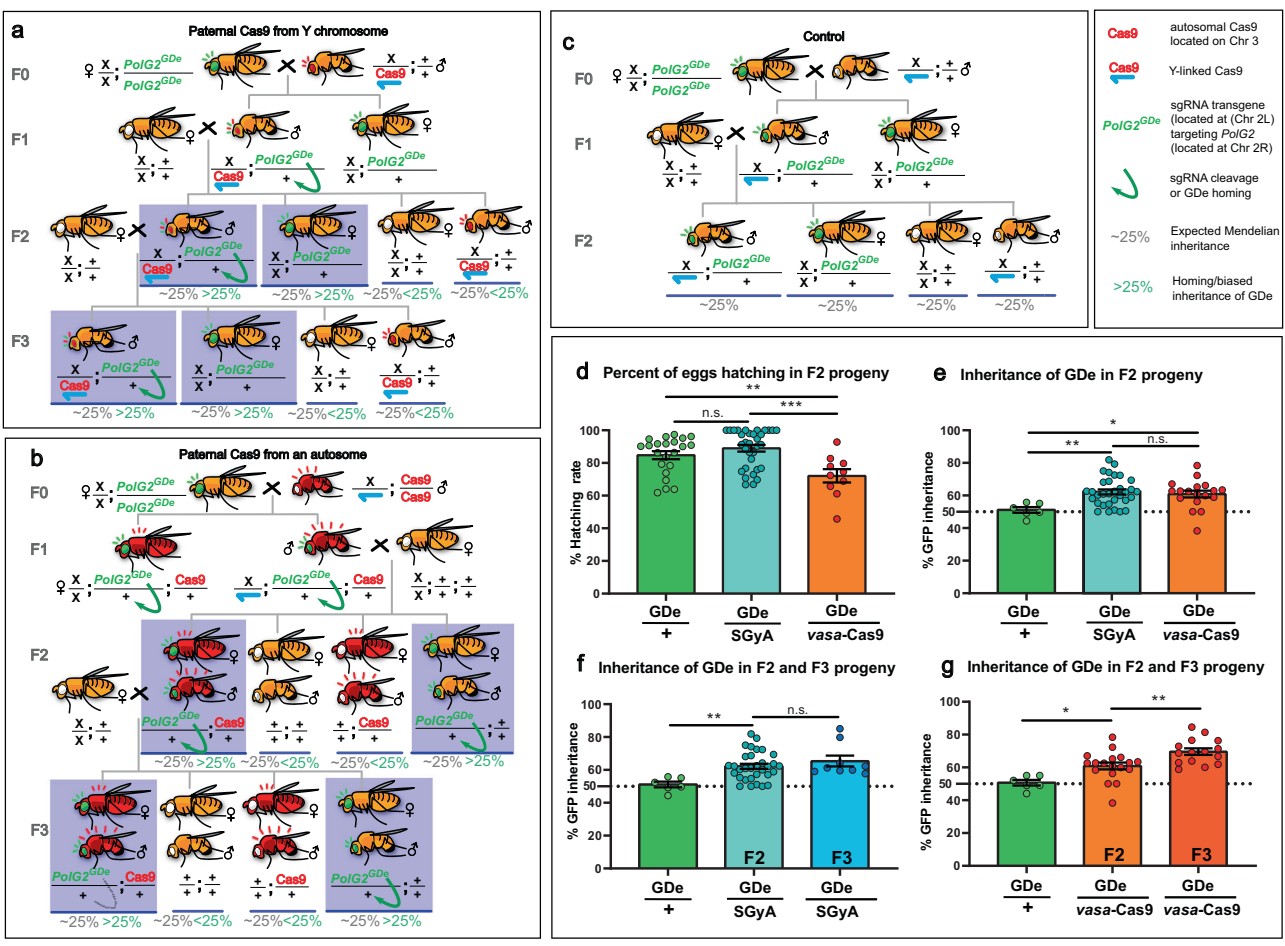

**Fig. 4 Functionality of SGyA as a split gene drive using a re-coded GDe. a** The crossing schematic involves the paternal Cas9 from the Y chromosome. The Cas9 transgene is only passed through the male germline. A SGyA male is crossed to homozygous GDe females to produce transheterozygous males and GDe-only females. The F1 transheterozygous male is then outcrossed to a w- females to assess GDe inheritance in F2 progeny. **b** The crossing schematic involves the paternal Cas9 from an autosome. A homozygous male harboring Cas9 on an autosome is outcrossed to a homozygous GDe female. Transheterozygous F1 males are subsequently outcrossed to w- females to assess GDe inheritance. Both sexes are affected. **c** Crossing schematic of the negative control cross. **d** Percent of F2 eggs hatched in crosses involving the control, transheterozygous SGyA males and transheterozygous autosomal Cas9 males. **e** Inheritance of the GDe in F2 progeny among different sources of Cas9. **f** Inheritance of the GDe in F2 and F3 progeny from crosses involving a transheterozygote male containing the GDe and the SGyA transgene. No significant deviations between both the F2 and F3 data sets were found (p value = 0.3446). **g** Inheritance of the GDe in F2 and F3 progeny from crosses involving a transheterozygote male containing the GDe and the autosomal Cas9 transgene. Significant differences between GDe inheritance observed in F2 progeny (when compared to control; p value = 0.0129) and GDe inheritance between F2 and F3 progeny (p value = 0.0051) Blue shaded boxes in crossing schematics highlight instances where a bias of GDe transmission is observed. Green arrows indicate the conversion of WT PolG2 allele into the GDe. Gray numbers represent the expected Mendelian inheritance percentages of the GDe. Green percentages indicate the homing/bias of GDe. At least 18 experimental/replicate crosses were set up per Cas9 experiment, and 6 for the control to determine F2 progeny outcomes. For F3 outcomes, at least nine experimental crosses were performed. Significance was determined using a two-tailed unpaired student's t test. For inheritance and male bias plots, vertical bars represent SEM. n.s. represents Non-significant. ***p < 0.0005; **p < 0.001; *p < 0.01. Source data is provided as a Source Data file.

Cas9 endonuclease from the Y chromosome in insects. Using this tool, we observed mutant phenotypes for a total of eight gene targets in the F1 offspring of crosses involving females carrying either tRNA-gRNA[32], or single gRNAs with males from either the autosomal- or Y-linked *vasa*-Cas9 lines. Crosses with the SGyA line resulted in mutant phenotypes only in the males, consistent with the Y-linked nature of this transgene. The Y-linked results also demonstrate somatic activity in the F1 male progeny which is consistent with previous studies showing that *vasa*-driven Cas9 expression is active in the soma of the zygote[18,43,44]. In contrast to a previous study[32], our Y-linked experiments with the tRNA-gRNA system did not result in high penetrance of mutant phenotypes, especially the lack of mutant phenotypes for the *cu* target. This difference may be explained by the PEV of the SGyA

transgene which can affect the levels of Cas9 expression. This is in contrast to the ubiquitous *actin5C* promoter used in the original tRNA-gRNA study and our autosomal cas9 experiments. In experiments involving SGyA and the sgRNAs targeting five phenotypic genes, we observed sex-dependent cleavage of *ct, tw, sc, wg,* and *ap* genes leading to phenotypic mutations (*ap, tw, sc*) or lethality (*ct* and *wg*). The mutant phenotypes observed in our study correlate with those reported for known mutant alleles of these genes[45–50].

The ability to sex sort is a crucial process for insect genetic control strategies including the Sterile Insect Technique[51–55]. Current genetic sexing systems (GSS) use widely applicable sexing approaches such as sex-specific phenotypes and genetic engineering approaches (using sex-specific introns) to facilitate sex

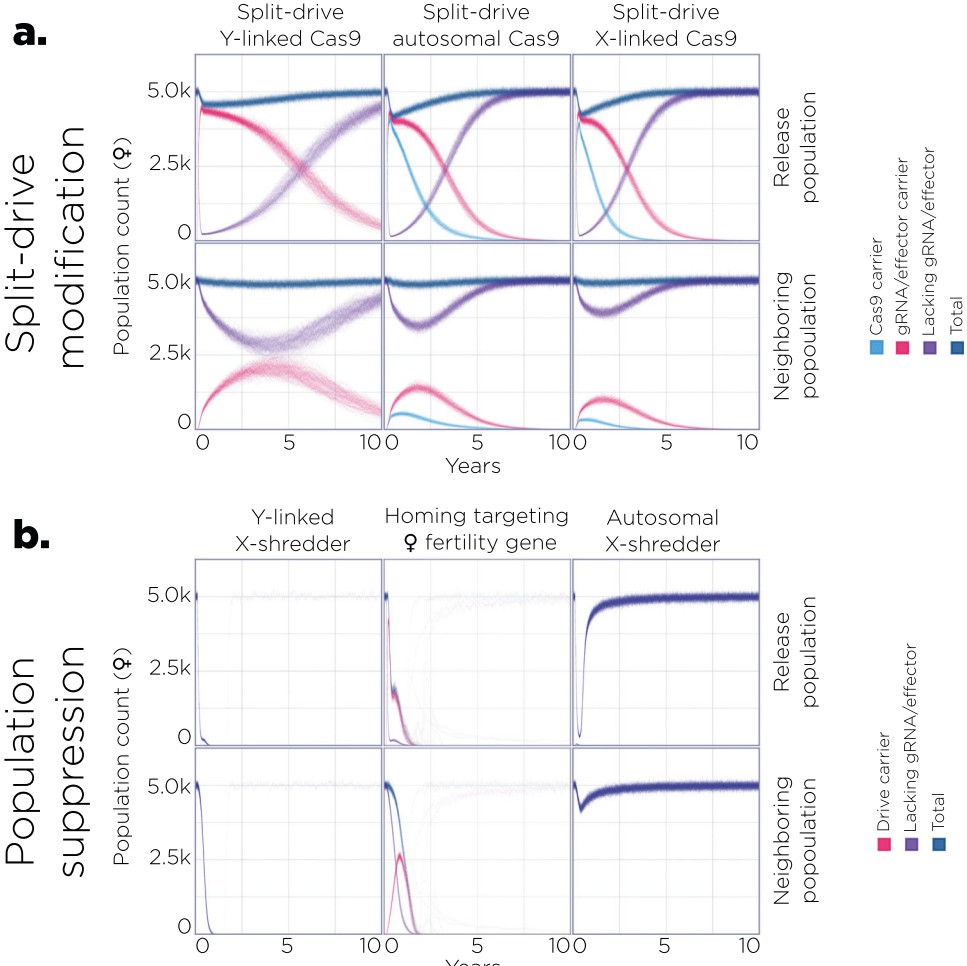

**Fig. 5 Population dynamics of SGyA-based gene drive systems. a** Model predictions for releases of *An. gambiae* mosquitoes homozygous or hemizygous for three different split drive systems intended for population modification. The SGyA-based system in which the Cas9 allele is Y-linked (left) is compared with an autosomal split drive system (middle) and a system in which the Cas9 allele is X-linked (right). In all cases, the gRNA/effector allele is autosomal. Life-history and gene drive parameters are provided in Supplementary Data File 4. 12 weekly releases were simulated in a population with an equilibrium size of 10,000 adults and a 1% per mosquito per generation migration rate with a neighboring population of the same equilibrium size. Model predictions were computed using 100 stochastic realizations of the MGDrivE framework[35]. Total adult female population size (dark blue), adult females carrying at least one copy of the gRNA/effector allele (red), adult females without the gRNA/effector allele (purple), and adult females carrying at least one copy of the Cas9 allele (light blue) were plotted for each system. Notably, for the Y-linked split drive system, the gRNA/effector allele persists longer in the population than for the autosomal or X-linked split drive systems. The Y-linked system also spreads to a higher frequency in the neighboring population. **b** Model predictions for equivalent releases of three population suppression systems: an SGyA-based Y-linked X-shredder (left), an autosomal homing-based drive targeting a gene required for female fertility (middle), and an autosomal X-shredder (right). Simulations assumed high rates of DNA cleavage and low rates of resistant allele generation, as required for effective population suppression[40] (Supplementary Data File 4). Total adult female population size (dark blue), adult females carrying at least one copy of the intact drive allele (red), and adult females without the intact drive allele (purple) were plotted for each system. Both the Y-linked X-shredder and autosomal homing-based drive targeting a female fertility gene achieved population elimination in >97% of simulations. The autosomal X-shredder leads to transient population suppression at the release site and limited spread to the neighboring population.

sorting in various insects[56–61]. We demonstrate that the SGyA line can be used as a GSS when coupled with a sgRNA that results in lethality after gene disruption, in a novel technique we term SELECT. Specifically, targeting the *PolG2* gene using the SGyA line produced LBM exclusively in males and resulted in 98% of males not surviving. While lethality was not 100% for *PolG2*, we did observe 100% male-specific lethality when targeting other genes such as *wg*, and *ct* indicating that penetrance of male-specific lethality may be gene-specific. Moreover, our results showed that an X-linked Cas9 is the complementary system to eliminate 100% of F1 females and to produce males exclusively. The combination of both strains produces an efficient GSS to SELECT for either gender.

It is well known that in *D. melanogaster* the position of a gene within euchromatic/heterochromatic regions can contribute to regulating expression. This phenomenon, called position effect variegation (PEV)[62], may explain the patchy expression of the fluorescent marker often observed in the eyes, and the overall reduced expression of Cas9 protein from the SGyA ♂'s as compared to an autosomal source of Cas9. In addition to this local effect on gene expression, global effects on the Y chromosome including meiotic sex chromosome inactivation (MSCI)[63] and sex chromosome-wide transcriptional suppression may also be contributing to our observations regarding reduced gene expression in some SGyA ♂'s (Supplementary Figure 1). Despite reduced gene expression in some individuals, our SGyA line still generated

robust Cas9 expression in the germline that was capable of generating heritable mutations (Supplementary Figure 1C).

To measure the efficiency of SGyA to mediate gene drive, we performed gene drive experiments in a confinable split drive context using a previously described re-coded *PolG2* HomeR gene drive element[26]. Despite our transgene being on the heterochromatic Y chromosome, the SGyA line was able to drive a GDe in the ♂ germline. Moreover, when compared to drive experiments using an autosomal source of *vasa*-Cas9, the SGyA had similar GDe inheritance rates which suggest that a Y-linked source of Cas9 can function and produce similar results to that from an autosome. However, when compared to previously generated split drives using male-specific promoters[26], our SGyA system still needs further optimization. For example, the best performing male-specific Cas9 line (using the *exuL* promoter) resulted in a mean homing rate of 75% of the GDe[26], whereas our SGyA line resulted in 69%.

As such, optimization may require inserting our transgene into another location on the Y or using alternative insulators[64]. In either case, our SGyA line enables the male-only transmission of the Cas9 component in a split drive system which allows for expression and gene drive conversion only in males thereby preventing the accumulation of resistant alleles that can result from maternal transmission[8,18,26,37]. This feature is critical for male genes drive systems like X-shredders, or other approaches such as sex-linked genome editors, e.g., targeting female fertility genes[17].

CRISPR-based X-shredder gene drives offer promising alternative approaches for insect vector and pest control. To date, only autosomal X-shredders have been successfully generated in *An. gambiae* malaria mosquitoes and in fruit flies[9–12]. Despite their success at distorting sex ratios using CRISPR-Cas9 or I-PpoI, the X-shredder components were inserted on autosomes rather than Y chromosomes. This may be for several reasons, including ease of inserting a transgene on an autosome rather than a heterochromatic Y chromosome and/or the meiotic silencing/sex chromosome-wide transcriptional suppression effects of sex chromosomes on an inserted transgene limiting robust expression[65–67]. Despite this progress, autosomal X-shredders limit the efficacy of this type of gene drive because only 50% of the ♂ progeny will inherit the X-shredder components. As a result, the gene drive will be "self-limiting" and will not be able to propagate continuously within the target population[67]. This is less than ideal when compared to an X-shredder system where components are linked to the Y chromosome. In a Y-linked configuration, 100% of the ♂'s will inherit the components for shredding X chromosomes and lead to the gene drive spreading into subsequent generations. Our work lays the groundwork for building X-shredder gene drives in *Drosophila*. We overcome the difficulty of inserting Cas9 on the Y by using HDR and finding a suitable region on the Y to robustly express transgenes. It should be noted that the Vasa promoter is generally premeiotic and therefore will not be suitable for meiotic X-shredding to generate male-biased progenies[12]. That said, future efforts to characterize suitable meiotic promoters and also to circumvent MSCI acting against meiotic specific promoters should be undertaken as this will be a necessary cornerstone to successfully engineer the X-shredder gene drive once these are inserted in the Y (or X) chromosome.

We used mathematical modeling to explore the utility of inserting Cas9 and other drive components on the Y chromosome. First, we compared the performance of a split drive with an autosomal gRNA/effector allele and a Y-linked, X-linked, or autosomal Cas9 gene. This revealed that, for default parameters, the gRNA/effector allele persists in the population for longer for the Y-linked system, partly due to the fact that female fertility

costs are not manifested for Y-linked Cas9 alleles. This could be useful for achieving more enduring population protection from vector-borne disease transmission, for instance. As a consequence, the gRNA/effector allele for the Y-linked system also spreads to a higher frequency and persists for longer in neighboring populations. Although this could be seen as detrimental to confinement, all three split drive designs are self-limiting, and spread to neighboring populations is transient, which may be an essential safety feature for initial gene drive acceptance and field trials[68]. Next, we compared the performance of a Y-linked X-shredder to an autosomal homing-based drive targeting a gene required for ♀ fertility[8]. This revealed that the Y-linked X-shredder achieves population elimination more quickly than the autosomal homing-based design. While resistant alleles can lead to a population rebound for both systems, Y-linked X-shredders can be engineered to target multiple loci on the X chromosome simultaneously[10], providing a clear route to minimizing resistant allele generation. Thus, for both suppression and modification, there are clear use cases and benefits to inserting drive components on the Y chromosome.

Altogether, this work represents a proof-of-principle guide of how to design transgenes for insertion into the heterochromatic Y chromosome using CRISPR in *D. melanogaster*. The approach to developing the SGyA line here could be used to develop X-shredders or split gene drives for the control of pests with XY sex systems (e.g., *D. suzukii*). Other potential uses include, but are not limited to, the ♂-specific cleavage of other target genes not explored in this study, the robustness of the Y chromosome-Cas9 expression throughout time. Our work provides the first described Y-linked Cas9 transgene in *D. melanogaster* that can be maintained as a stock for the use of CRISPR-based genetic studies and the creation of a ♂-specific split gene drive.

## Methods

**Construct design and assembly.** The Y chromosome is a difficult genomic landscape to build useful transgenesis-based tools. This is largely due to its highly heterochromatic nature and its ability to lead to position effect variegation. To overcome these issues, we previously generated constructs that are able to insert site-specifically using CRISPR tools and reduce the effect of gene silencing with the addition of insulator fragments[25]. To build upon these constructs, we generated three new vectors, SGyA, SGyB, and SGyC. We first amplified a ~7.9 kb *vasa*-Cas9−T2A-GFP fragment from a previously characterized *vasa* construct[2] using primers SGyA-1F and SGyA-1R. Then, this fragment was cloned into an AscI digested AByG backbone (Addgene # 111083, containing left homology arms span from 666,900 to 667,710 and right homology arms span from 666,064 to 666,876) using the Gibson assembly method[69]. Vector SGyA is available from Addgene (#160292). To generate vector SGyB, the same ~7.9 kb vasa-Cas9−T2A-GFP amplicon was cloned into an AscI digested AByF backbone (Addgene # 111084). To generate vector SGyC, the same AByG backbone vector was digested with AscI. A ~7.7 kb ubiq-Cas9−T2A-GFP fragment from a previously characterized *ubiquitin* construct[2] using primers SGyA-2F and SGyA-1R was cloned into the AByG backbone. To generate DNA double-stranded breaks for vector incorporation, a sgRNA targeting the Y chromosome was in vitro transcribed with the MEGAscript T7 Transcription Kit (ThermoFisher Scientific, Waltham, MA, USA) using self-annealing primers, SGygRNA-F and SGygRNA-R. All primer sequences are listed in Supplementary Table 1.

**SGyA fly line generation.** Fly maintenance and crosses were performed under standard conditions at 25 °C. Rainbow Transgenics (Camarillo, CA, USA) carried out embryo injections for the SGyA, SGyB, and SGyC plasmids. Embryos were injected with a premixed solution of the construct, Cas9 protein (PNA Bio Inc., Newbury Park, CA, USA) and in vitro transcribed sgRNAs into the Nos-Cas9−attp2 strain (y[1] sc[*] v[1] sev[21]; P{y[+t7.7] v[+t1.8]=nos-Cas9.R}attP2; BDSC #78782). Injecting the premix into a Cas9 line was performed to increase the chances of HDR insertion in the pole cells of embryos. A transformant was obtained for SGyA. No transformants were obtained for SGyB and SGyC. The genetic source of *Nanos*-SpCas9 (marked with vermillion [*v*]) was removed from the SGyA population by outcrossing these transformants to WT ♀'s. The resulting progeny were then scored to collect transgenic ♂'s (marked with tdTomato) without the *v* marker. These collected ♂'s lacking *v* were then outcrossed to WT ♀'s for multiple generations to ensure *Nanos*-SpCas9 was indeed not present. A stock was established by outcrossing these ♂'s to virgin w[1118] ♀'s. The SGyA

transgenic line (w[*]/TI{Disc\RFP[tdTom.3xP3]=vas-Cas9.T2A.GFP,attP}SGyA) is available from the Bloomington Stock Center (#91386).

**Fly lines to study SGyA efficiency to generate somatic mutations, to produce ♂ biased offspring and to function as a sex-biased split drive**. The transgenic fly line contains four gRNAs in a U6:3-tRNA-sgRNA expression system that targets *ebony* (*e*), *forked* (*f*), *sepia* (*se*), and *curled* (*cu*) was obtained from a previous publication[32]. The multiplexed gRNA transgene was inserted into the second chromosome (P{y[+t7.7]CaryP}attP40). In addition, we took advantage of the TRiP-CRISPR Knockout database (TRiP-KO; https://fgr.hms.harvard.edu/trip-knockout) and obtained five lines expressing sgRNAs targeting the genes *wingless* (BDSC# 81980), *cut* (BDSC#81942), *apterous* (BDSC#80345), *twisted* (BDSC#76991) and *scalloped* (BDSC#77055). All five gRNA transgenes were inserted on Chr 2, 25C6, 2 L:5108448. Finally, transgenic fly lines harboring a single gRNA targeting *PolG2* (U6.3-gRNA#1^PolG2; located on the second chromosome at site 8621) and a re-coded *PolG2* gene drive (GDe) (located on the 2nd chromosome in the *PolG2* gene) element were both obtained from a previous publication[26]. Lines were kept in an insect incubator at 25 °C, 50% HR, and 12:12 light:dark phase. For autosomal and sex chromosome comparisons, we also used two previously characterized *vasa*-Cas9 fly lines with a *vasa*-Cas9 transgene located on the third chromosome (86Fa)[2] and on the X chromosome in the *yellow* locus[70]. The autosomal *vasa*-Cas9 line is referred to as "autosomal Cas9" throughout the text and contains an Opie2-dsRED marker. The X-linked *vasa*-Cas9 line contains a 3xP3-dsRED marker.

**Quantification of differential gene expression from Y-linked and autotsomal-linked Cas9 lines**. Total RNA was extracted from the whole bodies of autosomal, SGyA, and WT adult 3–4 day old ♂'s using the miRNeasy Mini kit (Qiagen #217004). A total of 10 whole bodies were done per replicate for a total of 3 replicates for each type of sample. Following extraction, the RNA was treated with Ambion Turbo DNase (ThermoFisher Scientific #AM2238). The RNA quality was assessed using an RNA 6000 Pico Kit for Bioanalyzer (Agilent Technologies #5067-1513) and a NanoDrop 1000 UV-vis spectrophotometer (NanoDrop Technologies/Thermo Scientific, Wilmington, DE). mRNA was isolated using an NEBNext Poly(A) mRNA Magnetic Isolation Module (NEB #E7490), and libraries were constructed using an NEBNext Ultra II RNA Library Prep Kit for Illumina (NEB #E7770). The libraries were quantified using a Qubit dsDNA HS Kit (Thermo-Fisher Scientific #Q32854) and a High Sensitivity DNA Kit for Bioanalyzer (Agilent Technologies #5067-4626) and sequenced on an Illumina HiSeq2500 in single-read mode with a read length of 50 nt and sequencing depth of 30 million reads per library following the manufacturer's instructions. Reads were mapped to the *D. melanogaster* genome (Dmel Release 6) supplemented with the Tdtomato, dsRED, eGFP, and Cas9 sequences using STAR aligner 71, and the expression levels were determined with featureCounts 72 (Supplementary Data File 2). Correlation coefficients of the transcripts-per-million (TPM) values between WT and transgenic animals were calculated in R[14] and plotted with ggplot2 (Supplementary Figure 3). Differential expression analysis between transgenic and WT samples was performed using DESeq2[71]. In conjunction with DESeq2, the resulting genes were analyzed for overrepresentation of GO terms using the GOstats R software package[72]. Only GO terms with *p* value < 0.05 were selected. All RNA-sequencing data are available for download at NCBI BioProject number PRJNA748400.

**Efficient sex selection by exploiting sex chromosome-linked Cas9**. We evaluated the capabilities of the SGyA line to selectively kill ♂'s by using a gRNA expressing line targeting the essential gene polymerase gamma 35 K (*PolG2*). For each replicate, we prepared crosses of one ♀ from the line U6.3-gRNA#1^PolG2 to two ♂'s from the Y-linked Cas9, X-linked Cas9, or the autosomal *vasa*-Cas9 (10 replicates each). As a control, we mated the ♀'s to w[1118] ♂'s. We counted the F1 ♂'s and ♀'s from each cross and compared them among each treatment. Additionally, since our SGyA line males showed a wide variation of the fluorescent red marker expression, including males with no marker expression at all, we wanted to test whether the activity of Cas9 could have been impaired on those males. Thus, we independently crossed males with weak or moderate expression of the marker to homozygous U6.3-gRNA#1^PolG2 ♀'s (five replicates each) and estimated Cas9 activity based on the frequency of male F1 progeny that emerged.

**Characterization of the Y-encoded *vasa*-Cas9 fly line as part of a sex-biased split drive**. To test the functionality of our Cas9 transgene expressed from the Y chromosome (Y-Cas9), each replicate consisting of one transgenic Y-Cas9 ♂ was crossed to one homozygous transgenic virgin ♀ containing the GDe. A total of five individual crosses were set up. Progeny resulting from these crosses was scored for fluorescence (genotype) and sex. As the Y-Cas9 transgene is inherited through the Y chromosome of ♂'s, only transheterozygous ♂'s (containing Y-Cas9 and gene drive element) were obtained. Single transheterozygous ♂'s were then outcrossed to virgin w[1118] flies to observe functionality and drive dynamics of the gene drive element.

**Molecular characterization of flies and genotyping targeted loci**. To confirm the correct insertion of transgenes, PCRs were done on the genomic DNA of

transgenic flies and sequenced. Single-fly genomic DNA preparations were prepared by homogenizing a fly in 30μl of a freshly prepared squishing buffer composed of 10 mM Tris-Cl pH 8.0, 1 mM EDTA, 25 mM NaCl, and 200 μg/mL Proteinase K. Samples were then incubated at 37 C for 35 min and heated at 95 C for 2 min. Only 1.5 μl of genomic DNA was used as a template in a 50 μl PCR reaction with Q5 DNA polymerase (NEB). Primers amplifying the left and right boundaries of the SGyA transgene, SGyA-2F/SGyA-2R, and SGyA-3F/SGyA-3, respectively, were used to confirm the presence of the transgene in SGyA ♂'s. Primers GDe.1 F/GDe.3 R were used to confirm the right side of the re-coded gene drive element of *PolG2* (~3 kb) insertion. Primer pairs GDe.1 F/GDe.2 R were used to characterize mutations caused by NHEJ events in flies containing Cas9 and GDe (~500 bp). To sequence mutations in *se*, *e*, *f*, and *cu* target sites we used the following primer pairs: se-F/se-R for *sepia*, e-F and e-R for *ebony*, f-F and f-R for *forked*, and cu-F and cu-R for *curled*. Similarly, primers for amplifying *wingless*, *cut*, *apterous*, *twisted*, and *scalloped* were designed and tested. Retrogen Inc. performed the sequencing. All primers used can be found in Supplementary Table 1.

**Fly imaging**. Flies were scored and imaged on the Leica M165FC fluorescent stereomicroscope equipped with the Leica DMC2900 camera.

**Mathematical modeling**. We modeled the expected performance of Y-linked drive systems using the MGDrivE simulation framework[35] (https://marshalllab.github.io/MGDrivE/). This framework models the egg, larval, pupal, and adult life stages (both female and male adults are modeled) implementing a daily time step, overlapping generations, larval mortality that increases as a function of larval density, and a mating structure in which ♀ insects retain the genetic material of the adult ♂ with whom they mate for the duration of their adult lifespan. Although the Y-linked drive components described in this paper were engineered in *D. melanogaster*, we performed simulations for *An. gambiae* (life table parameters in Supplementary Data File 4) as a mosquito disease vector that proof-of-concept gene-editing tools from *D. melanogaster* could be applied to. We implemented the stochastic version of the MGDrivE framework to capture the randomness associated with low genotype frequencies and rare events such as resistant allele generation under some parameterizations. In this implementation, survival probabilities follow a binomial distribution, female mate choice follows a multinomial distribution over ♂ genotype frequencies accounting for mating competitiveness, daily offspring production follows a Poisson distribution with mean given by the genotype-specific fecundity of the adult ♀, and offspring genotype follows a multinomial distribution given the parental genotypes and inheritance pattern. 100 model repetitions were used for plots and statistics.

Two Y-linked drive systems were modeled: (i) a split drive system in which the Cas9 allele is Y-linked and the gRNA allele is autosomal[37], and (ii) a Y-linked X-shredder system in which the Cas9 and gRNA alleles are at a locus on the Y chromosome. Comparative modeling was performed in both cases. Versions of the split drive system were also modeled with: (i) the Cas9 allele at an unlinked autosomal locus, and (ii) the Cas9 allele being X-linked. Fitness costs for all split drive systems were derived from the expression of Cas9 (dominant, additive), effector gene expression at the gRNA locus (dominant, additive), and out-of-frame or otherwise costly (B) resistant alleles (dominant, additive). These fitness costs were applied to both ♀ fecundity and adult longevity. Two other population suppression systems were also modeled: (i) an autosomal X-shredder system, and (ii) an autosomal homing-based drive system targeting a gene required for female fertility[8]. For the Y-linked and autosomal X-shredder systems, fitness costs were derived from Cas9 expression (dominant, additive) and B-resistant alleles (dominant, additive). As X-shredding induces a fertility reduction, additional costs were only applied to adult longevity for this system. For the homing drive targeting a ♀ fertility gene, equivalent fitness costs were applied to adult longevity, and ♀ mosquitoes homozygous for the homing and/or B allele were considered infertile. All simulations were performed using the MGDrivE package (version 1.6.0) in R (https://www.r-project.org/) and analyzed using the MoNeT Python package (https://pypi.org/project/MoNeT-MGDrivE). Complete model and intervention parameters are listed in Supplementary Data File 4. Code is available upon request.

**Statistical analysis**. In all experiments, a minimum of three replicates was used to make comparisons between means. Data were subjected to Kolmogorov–Smirnov, Shapiro–Wilk, and Llieford tests to confirm a normal distribution. Due to the data presenting a normal distribution, we ran a Student's *t* test or a one-way ANOVA followed by a post hoc Tukey's test for multiple comparisons. Comparisons were considered statistically significant with *p* < 0.05. The software used for these analyses was GraphPad Prism version 8.3.1 for macOS (GraphPad Software, San Diego).

**Reporting summary**. Further information on research design is available in the Nature Research Reporting Summary linked to this article.

## Data availability

Complete sequence and plasmid DNA for vector SGyA are available at Addgene (#160292). All RNA-sequencing data are available for download at NCBI with BioProject: PRJNA748400. Source data are provided with this paper.

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

## Acknowledgements

This work was supported in part by funding from a DARPA Safe Genes Program Grant (HR0011-17-2-0047) and an NIH award (R01AI151004) to O.S.A., and an award from the Innovative Genomics Institute at UC Berkeley to J.M.M.

## Author contributions

O.S.A. conceived and designed experiments. S.G., D.C.R., S.C.M., and N.K. obtained genetic cross data; I.A. performed and analyzed RNA-Seq experiments; A.B. designed and T.Y. generated the SGyA, SGyB, and SGyC constructs; S.G., D.C.R., and J.E.D. performed molecular analyses; J.B., H.M.S.C., and J.M.M. performed mathematical modeling; O.S.A., S.G., and D.C.R. analyzed all the data. All authors contributed to writing and approved the final manuscript.

## Competing interests

O.S.A. is a founder of Agragene, Inc., and has an equity interest. The terms of this arrangement have been reviewed and approved by the University of California, San Diego in accordance with its conflict of interest policies. S.G. is currently affiliated with Agragene Inc. A.B. is currently affiliated with Verily Life Sciences. All other authors declare no competing interests.

## Ethical conduct of research

We have complied with all relevant ethical regulations for animal testing and research and conformed to the UCSD institutionally approved biological use authorization protocol (BUA #R2401).

## Gene drive safety measures

All crosses using gene drive genetics were performed in accordance with a protocol approved by the Institutional Biosafety Committee at UCSD, in which full gene drive experiments were performed in a high-security ACL2 barrier facility and split drive experiments were performed in an ACL1 insectary in plastic vials that were autoclaved prior to being discarded, in accordance with currently suggested guidelines for the laboratory confinement of gene drive systems[73,74].
