## [Peer Review File · Nature Communications]

Reviewers' Comments:

Reviewer #1:

Remarks to the Author:

The manuscript describes the action of a Y linked gene expressing the Cas9 endonuclease under the control of the vasa promoter with the experiments being performed in *Drosophila melanogaster*. The endonuclease was fused via a T2A self cleavage sequence to GFP gene which enabled detection and tracking of the transgene. The initial integration was achieved by homology-directed repair (HDR) facilitated by homology arms specific for Y chromosome sequence in a strain expressing Cas9 under Nanos promoter control to facilitate integration into germline chromosomes. The method here is also a little confusing since the authors state that they also injected "sgRNA complexed with recombinant SpCas9 protein" into the this Nanos-SpCas9 strain suggesting that there were two sources of Cas9 in these experiments. Perhaps this is an error in writing or maybe they were simply increasing their chances of integration via HDR. Either way this needs to be clarified. If the later then they should explain their logic. The rationale for the entire experiment was to show that Y-linked Cas9 expression would confine activity of the CRISPR/Cas9 system to only males in those insect species in which the sex determination system in XX/XY. In the application of genetic control technologies to insect pest control this includes many insect pests, for example many anopheline mosquitoes, many tephritid flies, and many calliphorid flies. The impact of male specific elimination/expression can therefore have a considerable impact on the development and application of new genetic control strategies.

The authors do show male specific activity of the single Y-integrated line they recovered. This is impressive but a major concern I have is that obtained a single line and the behavior of this line is the basis for this paper. I concede the difficulty in generating this line - integration into, and then expression from, the Y chromosome is problematic but this is one reason why they embarked on this experimental plan since achieving this is significant. Given they are working with the easiest system in insects, this being *Drosophila melanogaster*, my feeling is that they should have generated at least one other independent line at this location and/or used HDR to attempt integration at a closely adjacent site (i.e. another sgRNA target site in the same locus) or at another site in the Y. They do report decrease in the level of Cas9 expression (perhaps to be expected) but the significance is diminished by the use of only this single strain.

In terms of their modeling, I do not understand why they chose *Aedes aegypti*. This mosquito does not have an XX/XY system of sex determination, rather it has a sex locus. In contrast *Anopheles gambiae*, *An. coluzzii* and *An. stephensi* each possess an XX/XY sex determination system with the former two being the subject of a considerable program aimed at using gene drive to effect genetic control of these species. I would think these would have been better models. Indeed in the third paragraph of this section the authors discuss Y-linked gene shredders and reference *Anopheles*. Is the earlier reference to *Aedes* simply an error in writing? In some other aspects (e.g. spelling errors) the paper may have been hastily assembled and perhaps this is the cause here. I can't see why modeling could not also have developed for tephritid pests which are XY and, in the case of medfly, an X shredding technology has been demonstrated.

Minor points with figures:

Fig. 1E - what are the sizes of the expected fragments?

Fig.2A and text - can the explain the reason for including tRNA genes between the sgRNA sequences?

Fig. 2D - these photos are too small. Frankly I have trouble detecting the phenotypes that describe (e.g. ebony).

In summary, while I do not dispute the data and the significance, I am uncomfortable with all of it hinging on a single mutant. Put another way, if the authors had have obtained a single mutant not showing the expected behavior of a Cas9 gene being expressed on the Y, would they instead have sought more mutants? As described above I am left with the impression that the manuscript has been assembled rather quickly.

As such I cannot recommend acceptance at this stage.

Reviewer #2:

Remarks to the Author:

In this study Gamez and co-authors describe the generation of a *Drosophila* transgenic strain expressing the Cas9 protein under the *vasa* germline promoter from the Y chromosome. The authors present a detailed validation of this construct with different combinations of gRNA targeting a various gene. Although with some limitations derived by reduced activity in some of the conditions tested, the authors highlight the value of the Y-linked *vasa*-Cas9 line to expedite genetic sexing or to develop genetic control approaches with reduced potential to spread throughout populations compared to other gene drives thereby mitigating some of the downstream risks.

Main comments

The work is of clear value, however some of the sections and relative claims are not always clear or convincing to warrant publication of the current version of the manuscript in *Nat Comms*.

The efficacy of the system to induce sex-specific lethality seems clear, however the impact on the overall output or additional fitness costs associated should be clearer or addressed. This will be of relevance in comparison to other methods of sex separation prior release and accounted for in this manuscript.

The proof-of-concept of use to generate split gene drives is shown through a 3-generation assay. The supporting modeling show the effect over multiple generations however it is difficult to judge if the current constructs and strains would align to such dynamics and long-term expectations without experimental validation over multiple generations. I would imagine that the rathe of resistance generated would also play a role in these dynamics.

I would suggest the authors to revisit some of the specific sections, particularly some of the claims (see specific comments for 433-448). See specific comments for further recommendations which I hope the authors will find of use.

Specific comments

39 *D. melanogaster* - *Drosophila melanogaster*

43 Taken togehter - together

73 "even within populations 14" – meaning not fully clear without further info

77 relevant literature were such strategies were initially proposed should be included: see 10.1098/rspb.2002.2319, <https://doi.org/10.1098/rstb.2013.0432>, <https://doi.org/10.1098/rspb.2018.0776>

81 "...for limiting undesired activity in the maternal germline that can adversely affect spread of gene drives by generating DNA cleavage in the female germline known to damage paternal alleles 2,16–19" - the concept of cleave/rescue approach seems mixed with the germline carryover and zygotic deposition of nucleases known to bias NHEJ which can be source of functional resistance. The authors should eventually also refer to relevant literature if referring to the latter, see <https://doi.org/10.1371/journal.pgen.1007039>, <https://doi.org/10.1534/genetics.119.302037>, <https://doi.org/10.1534/genetics.116.197285>.

88-89 of using sex chromosomes for insect control is the development of synthetic sex ratio distorters (SRDs)¹³ - 10.1098/rspb.2002.2319 and other pioneering works (e.g., Hamilton etc) seems more appropriate here?

92-93 "resulting in biased transmission of X-chromosome bearing gametes 10–12,20,21" – ref 21

does not show experimental data on sex distortion therefore not relevant to the statement.

93-95 It is unclear to me what is intended for "necessary for making them truly applicable to the insect genetic control designs" and how reference

12 is of support for the statement. Reference 9 to 12 provide plenty of evidence that self-limiting or self-sustaining approaches based on SRD can be engineered without requiring Y-linkage.

97 "predicted to be one of the most rapid and resilient gene drive strategies 22" – if intentionally referring to CRISPR-based only ref 10 should be added. If intended to X-shredding in general ref 35 and 11 should be included here.

103 "have recently established a method to insert transgenes on the *Drosophila melanogaster* Y chromosome 23." – considering the heavily mosquito-related literature utilised the equivalent method developed in *Anopheles* seems relevant here (current ref 40)

125 It is unclear from this sentence if the construct was generated as part of this work or in ref 19. If the latter the original work relevant to *vasa* characterisation seems appropriate here.

127 "we incorporated a marker to assess promoter activity as done previously 19,26." – again the comparable work and Y-linkage of a *vasa*-gfp marker in *Anopheles* Y (ref 40) seems more appropriate here

145-146 Information regarding the choice of target site on the Y chromosome should be included. It would be interesting to know if the position of insertion may be relevant for the variegated expression pattern.

150 "We therefore maintained the SGyA line by selecting σ individuals with relatively high levels of marker expression" The writing here seems slightly misleading as results in 1C do not seem to require selection of individuals with high expression. Also it will be interesting to know if the variegated phenotype is reduced or eliminated by selecting for strong expression. Probably not if the phenotype is caused by heterochromatic silencing. Additional info regarding the KO/phenotype used in S1C should be included in the main.

163-181 Not fully clear with the justification for performing a full RNA seq to measure the Cas9 expression. I would imagine that qPCR and western blot from testes may have served for the purpose and possibly given a better quantification of the Cas9 protein produced? With the current background provided is also difficult to understand the real meaning of the downstream differential expression comparisons. E.g. how many individuals were used for the sequencing? What tissues were used? What kind of extraction and seq procedure was applied? Considering the variability of gRNA activity shown in the following section, correct expression of gRNA may have been useful if captured. The conclusion of this sections also appears quite vague as this was already confirmed in the previous section.

234-235 Considering the variability among gRNAs and aut vs Y-linked shown throughout the extensive validation performed the conclusion seems slightly simplistic. Effectively the following "Taken together, these data suggest our SGyA line is able to efficiently produce mutant phenotypes in a σ specific manner" was already shown in S1C described earlier on.

347-370 This section may be of interest although not clear how related to the current work, which is not showing the engineering of any of these technologies. The modeling and the corresponding engineering was already shown elsewhere therefore should give further justification for including this section here. i.e. a closer comparison and discussion between these self-sustaining and the limiting technologies including the one shown here rather than focusing on comparisons among self-sustaining approaches which seems less relevant to this work.

386 results for the *cu* target do not seem included in Fig 2

387 PEV should be Position Effect Variegation (PEV) of the gRNA transgene as the SGyA is the same for all the gRNA constructs. Info regarding method and position of integration of gRNA

should be included in the relative section of result.

425 More info are required for readers to have a better idea of the comparison to previous systems (developed in 24)

433-448 This section appears rather misleading as it is. The insertion of transgenes into the Y chromosome have been shown previously, by the same authors in *Drosophila* and by other groups in mosquitoes with the generation of Y-linked docking lines. The problem of meiotic silencing has been evidenced in *Anopheles* in several studies evidencing the impact of meiotic inactivation on X and Y chromosome (see <https://doi.org/10.1038/s41598-019-51181-1>, <https://doi.org/10.1186/1471-2148-12-69>, ref 14, and <https://www.biorxiv.org/content/10.1101/860551v2.abstract> etc.)

REVIEWER COMMENTS (Author responses in BLUE)

Reviewer #1 (Remarks to the Author):

The manuscript describes the action of a Y linked gene expressing the Cas9 endonuclease under the control of the vasa promoter with the experiments being performed in *Drosophila melanogaster*. The endonuclease was fused via a T2A self cleavage sequence to GFP gene which enabled detection and tracking of the transgene. The initial integration was achieved by homology-directed repair (HDR) facilitated by homology arms specific for Y chromosome sequence in a strain expressing Cas9 under Nanos promoter control to facilitate integration into germline chromosomes. The method here is also a little confusing since the authors state that they also injected "sgRNA complexed with recombinant SpCas9 protein" into the this Nanos-SpCas9 strain suggesting that there were two sources of Cas9 in these experiments. Perhaps this is an error in writing or maybe they were simply increasing their chances of integration via HDR. Either way this needs to be clarified. If the later then they should explain their logic. The rationale for the entire experiment was to show that Y-linked Cas9 expression would confine activity of the CRISPR/Cas9 system to only males in those insect species in which the sex determination system is XX/XY. In the application of genetic control technologies to insect pest control this includes many insect pests, for example many anopheline mosquitoes, many tephritid flies, and many calliphorid flies. The impact of male specific elimination/expression can therefore have a considerable impact on the development and application of new genetic control strategies.

Thank you for your constructive feedback. We apologize for not making the text clearer regarding our rationale for injecting our premix into a Cas9 line. We have added the following text in the methods section under SGyA fly line generation:

"Injecting the premix into a Cas9 line was performed to increase the chances of HDR insertion in the pole cells of embryos."

The authors do show male specific activity of the single Y-integrated line they recovered. This is impressive but a major concern I have is that obtained a single line and the behavior of this line is the basis for this paper. I concede the difficulty in generating this line - integration into, and then expression from, the Y chromosome is problematic but this is one reason why they embarked on this experimental plan since achieving this is significant. Given they are working with the easiest system in insects, this being *Drosophila melanogaster*, my feeling is that they should have generated at least one other independent line at this location and/or used HDR to attempt integration at a closely adjacent site (i.e. another sgRNA target site in the same locus) or at another site in the Y. They do report a decrease in the level of Cas9 expression (perhaps to be expected) but the significance is diminished by the use of only this single strain.

We agree that the generation of additional transgenic lines at different sites may help us and others understand known phenomena such as position effects and perhaps the meiotic

silencing effects of the Y chromosome. However, the scope of this work was to generate a transgenic Y-linked Cas9 line that was able to generate mutants efficiently and test its ability for genetic sexing and gene drive. We were really lucky enough to obtain a single transformant for the position spanning 666,877 through 666,899 on the Y. Although we did not mention this in the text, we had also generated a second similar construct with homology arms spanning a different but close Y chromosome position (666,513-666,535). Unfortunately, we were unable to obtain a transformant for this second line and thus we excluded this from the paper. In addition, we generated another construct with the same homology arms as SGyA but contained a Ubiquitin-Cas9 rather than Vasa-Cas9. This construct was also unsuccessful at generating transformants. We must point out that while *Drosophila melanogaster* is a well-studied model animal, this does not mean our task was easy. We should note that two factors likely played a role in the difficulty of obtaining a Y-linked Cas9. First, the efficiency of inserting a transgene by HDR is substantially low (this is well known in the gene editing field) and second, inserting a transgene via HDR in a highly heterochromatic chromosome is not simple due to the issue of accessibility. Knowing this, we understood that generating multiple lines would require significant time and effort to enable us to either reconfirm our current results or provide an unsuccessful example/line. In either case, we conclusively demonstrate that we can achieve Cas9 integration on the Y-Chromosome, this line is robustly expressed and is stable and can be used to generate mutants, sex selection and even gene drives - we believe this to be a significant accomplishment that conclusively paves the way for similar technologies in global pests such as the deadly malaria vector - *Anopheles gambiae* which has a sequenced Y-chromosome. The main purpose of this study is to create a useful tool for the community and to show it is possible to insert CRISPR transgene on the Y chromosome of Drosophilids. This work is a follow-up of Buchman and Akbari, 2018.

In terms of their modeling, I do not understand why they chose *Aedes aegypti*. This mosquito does not have an XX/XY system of sex determination, rather it has a sex locus. In contrast *Anopheles gambiae*, *An. coluzzii* and *An. stephensi* each possess an XX/XY sex determination system with the former two being the subject of a considerable program aimed at using gene drive to effect genetic control of these species. I would think these would have been better models. Indeed in the third paragraph of this section the authors discuss Y-linked gene shredders and reference *Anopheles*. Is the earlier reference to *Aedes* simply an error in writing? In some other aspects (e.g. spelling errors) the paper may have been hastily assembled and perhaps this is the cause here. I can't see why modeling could not also have developed for tephritid pests which are XY and, in the case of medfly, an X shredding technology has been demonstrated.

We thank the reviewer for this comment. We had chosen *Ae. aegypti* because a split-drive system has been engineered in that species, and because we have a DARPA program grant to engineer gene drives in *Ae. aegypti*. Also, the M locus could be thought of as equivalent to an XX/XY system. However, the reviewer is absolutely correct - *An. gambiae* is a much better choice for this paper as: i) it is a true XX/XY system, ii) there are several research programs to engineer gene drives in this species, and iii) we refer to the Y-linked X-shredders previously

developed in *Anopheles*. In response, and to remedy this, we have updated our life history parameters in Table S4 to correspond to *An. gambiae*, and have repeated all simulations, including the output in Figure 5, to represent *An. gambiae* instead of *Ae. aegypti*. All relevant sections of the text, including species mentions and simulation results, have been updated accordingly. Once again, we are grateful to the reviewer for pointing this out. This greatly improves the modeling portion of the paper. R.e. the question of why modeling results weren't also provided for the medfly or other tephritid pests, we felt that a single species was sufficient to demonstrate the comparative performance of modification and suppression gene drive systems with Y-linked components, also considering that this is primarily a molecular biology paper.

Minor points with figures:

Fig. 1E - what are the sizes of the expected fragments?

Corrected - We have updated this figure to outline primer pairs and expected sizes. We also added the following sentences in the figure legend to indicate expected band sizes:

"Expected band size for the left side primer pair is 1.690 kb. Expected band size for the right side primer pair is 1.893 kb."

LHA primers 1054A.5 - 842B (1690 bp)

RHA primers 1054A.3 and 941G.SQ2 (1893 bp)

Fig.2A and text - can the explain the reason for including tRNA genes between the sgRNA sequences?

Corrected - We have added a sentence to the 2A figure legend and to the main text to clarify this point.

2A Figure legend

"Flanking the gRNA's with tRNAs enables expression from a single promoter and processing of the multiplexed gRNAs."

Main text

"This experiment was performed to assess a previously tested multiplexed tRNA-gRNA system on SGyA which uses a strategy to flank the gRNA's with tRNAs to enable expression from a single promoter and processing of the multiplexed gRNAs³²."

Fig. 2D - these photos are too small. Frankly I have trouble detecting the phenotypes that describe (e.g. ebony).

Corrected - We apologize for the small images. To help see the mutations, we separated figure 2 into two separate figures. Figure 2 will not have the mutant images. Figure S4 will contain the images of the mutants in a larger format for visibility. As a result of this supplemental figure addition, we've changed the subsequent supplemental figure numberings throughout the text.

In summary, while I do not dispute the data and the significance, I am uncomfortable with all of it hinging on a single mutant. Put another way, if the authors had have obtained a single mutant not showing the expected behavior of a Cas9 gene being expressed on the Y, would they

instead have sought more mutants? As described above I am left with the impression that the manuscript has been assembled rather quickly.

As such I cannot recommend acceptance at this stage.

Thank you for your thorough review and assessment of the manuscript. As mentioned before, it is possible that we see the resource we are publishing here differently. To us, this is a roadmap on how to insert sequences onto Y chromosomes of insects, using *Drosophila* as a model, and a follow-up study of previous work we did to establish this target site on the Y chromosome. Here we have re-used our previous site, given the difficulty to create additional sites (information we have now added back to the manuscript), thus providing further confidence for the community regarding the availability and usability of this site. We neither claim to have generally improved our understanding of the biology of the Y, nor on the evolutionary forces (e.g. MSCI) acting upon it, for which, yes, more lines would be needed. Obtaining a second insertion line proved to be exceedingly difficult and since one additional line (following the initial insertion in Buchman et al) is sufficient to demonstrate proof of this concept we chose to move forward with the SGyA line. Obtaining an additional line would take more effort and time to reach a similar conclusion. The scope of the work is proof-of-concept to demonstrate that a Cas9 transgene was able to function on the Y chromosome.

Reviewer #2 (Remarks to the Author):

In this study Gamez and co-authors describe the generation of a *Drosophila* transgenic strain expressing the Cas9 protein under the vasa germline promoter from the Y chromosome. The authors present a detailed validation of this construct with different combinations of gRNA targeting a various gene. Although with some limitations derived by reduced activity in some of the conditions tested, the authors highlight the value of the Y-linked vasa-Cas9 line to expedite genetic sexing or to develop genetic control approaches with reduced potential to spread throughout populations compared to other gene drives thereby mitigating some of the downstream risks.

Main comments

The work is of clear value, however some of the sections and relative claims are not always clear or convincing enough to warrant publication of the current version of the manuscript in Nat Comms.

Thank you for this feedback. We have significantly improved the manuscript and hope that now it will be acceptable.

The efficacy of the system to induce sex-specific lethality seems clear, however the impact on the overall output or additional fitness costs associated should be clearer or addressed. This will be of relevance in comparison to other methods of sex separation prior release and accounted for in this manuscript.

We did not find any obvious fitness costs associated with fertility, fecundity or survivorship in our SGyA transgenic males. Since we did not perform any fitness experiments to compare both the autosomal Cas9 line and the Y-linked Cas9 line, we did not mention fitness. However, experiments to determine fitness cost will be ideal for a future study where the SGyA line is to be tested as an X-shredder gene drive. The current study aims to determine whether (1) it is feasible to generate a Y-linked Cas9 transgenic line, (2) able to stably express Cas9 for knock-out experiments, and (3) function as a split gene drive using a previously validated gene drive - all of which have been demonstrated. We must note that fitness of the transgenic insect will depend on the integration of the transgene and will thus need to be tested on a case-by-case basis. Testing fitness will be especially important when integrating a CRISPR transgene into the Y (or heterogametic) chromosome in an actual pest animal. Since our model organism is not an insect pest, determining the effect of transgene on fitness was not our main goal since this particular transgenic strain will not be used for insect control--it is a proof of concept.

While we do mention several times in the text that the SGyA line can be used for insect control, this is done as a proof of concept to demonstrate the importance and impact of this current work. We are excited to share this crucial piece of work because it is also useful for other research where a Y-linked Cas9 is desired. We understand that performing additional experiments (to determine fitness) is ideal, but we hope the experiments performed here are enough to warrant publication.

The proof-of-concept of use to generate split gene drives is shown through a 3-generation assay. The supporting modeling show the effect over multiple generations however it is difficult to judge if the current constructs and strains would align to such dynamics and long-term expectations without experimental validation over multiple generations. I would imagine that the rate of resistance generated would also play a role in these dynamics.

Indeed, performing additional outcrosses can give us some insight into the behavior dynamics of this split drive. We also feel that modeling is also helpful in that regard too. Like all other drive systems published, we really don't know how constructs/strains will behave over multiple generations in an applicable wild setting. Here we just want to demonstrate that our Y-linked Cas9 line can indeed bias mendelian inheritance - and we show that over successive generations this is indeed the case.

I would suggest the authors to revisit some of the specific sections, particularly some of the claims (see specific comments for 433-448). See specific comments for further recommendations which I hope the authors will find of use.

We thank the reviewer for their helpful comments.

Specific comments

39 D. melanogaster - Drosophila melanogaster
Correction applied.

43 Taken together - together
Correction applied.

73 “even within populations 14” – meaning not fully clear without further info
We removed this part of the sentence to make it clearer. Additional meaning/enhancement would not change the meaning of this sentence.

77 relevant literature were such strategies were initially proposed should be included: see
10.1098/rspb.2002.2319, <https://doi.org/10.1098/rstb.2013.0432>,
<https://doi.org/10.1098/rspb.2018.0776>
We updated our citations to include this literature.

81 “...for limiting undesired activity in the maternal germline that can adversely affect spread of gene drives by generating DNA cleavage in the female germline known to damage paternal alleles 2,16–19” - the concept of cleave/rescue approach seems mixed with the germline carryover and zygotic deposition of nucleases known to bias NHEJ which can be source of functional resistance. The authors should eventually also refer to relevant literature if referring to the latter, see <https://doi.org/10.1371/journal.pgen.1007039>, <https://doi.org/10.1534/genetics.119.302037>, <https://doi.org/10.1534/genetics.116.197285>.
We have updated the references to reflect the latter statement (functional resistant alleles). The sentence now reads:

“...or for limiting undesired activity in the maternal germline that can adversely affect spread of gene drives by generating functional resistant alleles in the female germline”

88-89 of using sex chromosomes for insect control is the development of synthetic sex ratio distorters (SRDs)¹³ - 10.1098/rspb.2002.2319 and other pioneering works (e.g., Hamilton etc) seems more appropriate here?
References have been added.

92-93 “resulting in biased transmission of X-chromosome bearing gametes 10–12,20,21” – ref 21 does not show experimental data on sex distortion therefore not relevant to the statement.
Citation removed.

93-95 It is unclear to me what is intended for “necessary for making them truly applicable to the insect genetic control designs” and how reference 12 is of support for the statement. Reference 9 to 12 provide plenty of evidence that self-limiting or self-sustaining approaches based on SRD can be engineered without requiring Y-linkage.
We apologize for the confusion. We modified the sentence to state the following:

“Although previously described SRDs^{10–12,23} are effective without being Y-linked, maintaining these SRD traits within a target insect population will require multiple releases of transgenic individuals, rendering this approach costly. On the other hand, SRDs which are Y-linked can reduce this cost considerably²⁴.”

97 “predicted to be one of the most rapid and resilient gene drive strategies 22” – if intentionally referring to CRISPR-based only ref 10 should be added. If intended to X-shredding in general ref 35 and 11 should be included here.

We have replaced the previous reference with reference 10 (Galizi et al., 2016).

103 “have recently established a method to insert transgenes on the *Drosophila melanogaster* Y chromosome 23.” – considering the heavily mosquito-related literature utilised the equivalent method developed in *Anopheles* seems relevant here (current ref 40)

We respectfully disagree with the reviewer. Reference 40 (Bernardini et al., 2014) is a study where the initial Y-chromosome integration of a transgene was obtained by transposon-mediated integration (and then yes followed by HDR). However, in this case we are citing our previous work (which pioneered CRISPR-HDR into an endogenous sequence of the Y and not one previously randomly inserted into). In any case, we apologize for the confusion and have modified it to state:

“...we have recently established a method to insert transgenes on the *Drosophila melanogaster* Y chromosome using CRISPR-based HDR insertion.”

125 It is unclear from this sentence if the construct was generated as part of this work or in ref The reference included here refers to the nature of *vasa* expression (present in germline and soma). The construct generated in this work, SGyA, was not previously made. We left the sentence as is.

19. If the latter the original work relevant to *vasa* characterisation seems appropriate here. Please see the response above.

127 “we incorporated a marker to assess promoter activity as done previously 19,26.” – again the comparable work and Y-linkage of a *vasa*-gfp marker in *Anopheles* Y (ref 40) seems more appropriate here
Appropriate reference added.

145-146 Information regarding the choice of target site on the Y chromosome should be included. It would be interesting to know if the position of insertion may be relevant for the variegated expression pattern.

The target site integration was chosen based on the previous article (Buchman and Akbari, 2019) where several integration sites were assessed. The insertion site that spanned from 666,900 to 666,876 on the Y chromosome had previously been successful. When attempted again in our current work using our Cas9 transgene, we also saw successful integration.

In our methods under the construct design and assembly section, we state that we built upon that previous construct to generate SGyA which is what this work is based on.

150 “We therefore maintained the SGyA line by selecting ♂ individuals with relatively high levels of marker expression” The writing here seems slightly misleading as results in 1C do not seem to require selection of individuals with high expression.

We have modified the sentence to make it clearer:

“We therefore maintained the SGyA line by allowing the ♂s with varied fluorescent marker expression to mate with WT ♀s each generation (Fig. S1B).”

Also it will be interesting to know if the variegated phenotype is reduced or eliminated by selecting for strong expression. Probably not if the phenotype is caused by heterochromatic silencing.

We did not observe disappearance of the variegated phenotype.

Additional info regarding the KO/phenotype used in S1C should be included in the main.

We added the following to the main text:

“...still displayed sufficient Cas9 activity that was capable of editing and generating high rates of lethality in males that inherited a sgRNA transgene targeting PolG2 (Fig. S1C). We therefore maintained the SGyA line by allowing the ♂s with varied fluorescent marker expression to mate with WT ♀s each generation (Fig. S1B).”

163-181 Not fully clear with the justification for performing a full RNA seq to measure the Cas9 expression. I would imagine that qPCR and western blot from testes may have served for the purpose and possibly given a better quantification of the Cas9 protein produced? With the current background provided, it is also difficult to understand the real meaning of the downstream differential expression comparisons. E.g. how many individuals were used for the sequencing? What tissues were used? What kind of extraction and seq procedure was applied? Considering the variability of gRNA activity shown in the following section, correct expression of gRNA may have been useful if captured. The conclusion of this sections also appears quite vague as this was already confirmed in the previous section.

We apologize for not making our intentions clearer in the text. While qPCR and western blots may have given good quantification, we believe RNA sequencing is the best choice to give accurate quantification of the transcript expression levels. In the methods section, we indicate which whole bodies of males were used for total RNA extraction and which sequencing procedure was applied. We also added the following in the methods text:

“A total of 10 whole bodies were done per replicate for a total of 3 replicates for each type of sample.”

234-235 Considering the variability among gRNAs and aut vs Y-linked shown throughout the extensive validation performed, the conclusion seems slightly simplistic. Effectively the following “Taken together, these data suggest our SGyA line is able to efficiently produce mutant phenotypes in a ♂ specific manner” was already shown in S1C described earlier on.

We believe this conclusion is appropriate for summarizing the overall result of these experiments. Adding additional information may not make the conclusion clearer than it needs to be.

347-370 This section may be of interest although not clear how related to the current work, which is not showing the engineering of any of these technologies. The modeling and the corresponding engineering was already shown elsewhere therefore should give further justification for including this section here. i.e. a closer comparison and discussion between

these self-sustaining and the limiting technologies including the one shown here rather than focusing on comparisons among self-sustaining approaches which seems less relevant to this work.

We believe that the mathematical modeling section plays a valuable role in the manuscript, as Y-encoded Cas9 is characterized as part of a split gene drive system in the experiments, and the potential use of Y-linked Cas9 in population suppression systems is described in the Abstract, Introduction and Discussion sections. Given the focus on Y-linked Cas9, we think it makes sense to explore any benefits that may have over X-linked or autosomal sources of Cas9, for both population replacement and suppression technologies. Although these technologies and modeling have been described elsewhere, other models have not focused on this comparison to the same extent. We agree with the reviewer that we could have provided more justification for including this section here, and have added the following text in the first sentence of the section:

“Advancing upon the characterization of Y-encoded Cas9 functioning as a split gene drive, and the goal of utilizing Y-encoded Cas9 as a population suppression system, we performed modeling to explore the potential for SGyA-based drive systems to enact efficient population modification and suppression.”

386 results for the *cu* target do not seem included in Fig 2

We apologize for not including in the text and figure legend that *cu* mutants were not recovered. Because of this, no data was presented for this particular mutant in figure 2. The following sentences were added in the main text and figure legend, respectively:

“No mutants were recovered for the *cu* target and were therefore excluded from the figure.”

“No *cu* mutants were obtained and were thus omitted.”

387 PEV should be Position Effect Variegation (PEV) of the gRNA transgene as the SGyA is the same for all the gRNA constructs. Info regarding method and position of integration of gRNA should be included in the relative section of the result.

We apologize for not including insertion site information for the gRNA transgenes tested. We have added this information as the following sentences:

“The multiplexed gRNA transgene was inserted into the second chromosome (P{y[+t7.7]CaryP}attP40).”

“All five gRNA transgenes were inserted on Chr 2, 25C6, 2L:5108448”

“Finally, transgenic fly lines harboring a single gRNA targeting PolG2 (U6.3-gRNA#1PolG2; located on the second chromosome at site 8621) and a re-coded PolG2 gene drive (GDe) (located on the 2nd chromosome in the PolG2 gene) element were both obtained from a previous publication 25.”

425 More info are required for readers to have a better idea of the comparison to previous systems (developed in 24)

We apologize for not including more information. We added the following sentence:

“For example, the best performing male-specific Cas9 line (using the *exuL* promoter) resulted in a mean homing rate of 75% of the GDe26, whereas our SGyA line resulted in 69%.”

433-448 This section appears rather misleading as it is. The insertion of transgenes into the Y chromosome have been shown previously, by the same authors in *Drosophila* and by other

groups in mosquitoes with the generation of Y-linked docking lines. The problem of meiotic silencing has been evidenced in Anopheles in several studies evidencing the impact of meiotic inactivation on X and Y chromosome (see <https://doi.org/10.1038/s41598-019-51181-1>, <https://doi.org/10.1186/1471-2148-12-69>, ref 14, and <https://www.biorxiv.org/content/10.1101/860551v2.abstract> etc.)

The reviewer is correct to state that insertion of transgenes into the Y chromosome has been shown previously (we state this in the first sentence of the discussion). However, in the referenced section (lines 433-448), we describe a lack of Y-linked CRISPR component X-shredders.

In reference to the second line of the reviewer's comment, we apologize for not referencing meiotic silencing in this section. We've added a sentence that states:

“...and/or the meiotic silencing effects of sex chromosomes on an inserted transgene limiting robust expression”

Reviewers' Comments:

Reviewer #1:

Remarks to the Author:

I am pleased that the authors removed the modeling of *Aedes aegypti* and replaced it with a species that contains a Y chromosome and their choice of *Anopheles gambiae* is perfect.

I am less pleased with their response about using only a single mutant upon which to base their analysis. I completely understand the profound difficulty of engineering the Y chromosome and I applaud their ability to achieve this. Logically speaking, they should be able to do so again, and perhaps a third time, but I do understand this will take time. From my perspective it remains the desired course.

In their rebuttal the authors point out that it will at least equally difficult to achieve similar modifications of the Y chromosome of pest insects but clearly this is the direction they wish to go and it is the rationale for the research. So I would consider this question from the perspective of readers who wish to follow their path. They would be best served by multiple mutants. If this cannot be achieved here then the least the authors can do is to include the description of their "failed" mutants (that is, those described in their rebuttal) in the paper itself so that these readers can get an estimate of the pitfalls ahead. I think this also serves the publisher well since it makes the paper more informative and useful from this practical sense.

My advice to the editor is that this inclusion is mandatory for publication.

Reviewer #2:

Remarks to the Author:

This is a scientifically sound and useful work and I believe that the manuscript will be of interest in the field as well as suitable for the Nature Communications audience. The current version seems reasonably improved, particularly in terms of clarity. There are only a few aspects that I would recommend the authors to clarify in the discussion and/or relevant sections of the manuscript before publication.

1. "In addition to this local effect on gene expression, a global effect of the Y chromosome termed meiotic sex chromosome inactivation 62, may also be playing a role. These phenomena are likely involved in our observations regarding reduced marker gene expression in some SGyA σ 's (Fig. S1)."

I believe that the authors here may want to refer to the sex chromosome-wide transcriptional suppression rather than meiotic silencing as the latter would have no effect (at least not on its own) on the expression of the 3xP3 driven RFP marker reported in S1 but only on the Cas9/GFP expressed in the germline.

2. "As such, optimization may require inserting our transgene in another location on the Y or using alternative insulators 63. In either case, our SGyA line enables the male-only transmission of the Cas9 component in a split drive system which allows for expression and gene drive conversion only in males thereby preventing the accumulation of resistant alleles that result from maternal transmission 8,18,26,36. This feature is critical for male-gene drive systems like X-shredders." There is little discussion or reference in previous sections about the utility of insulators as well as regarding the feasibility of targeting other Y loci. Probably useful to include in the supplementary some more information regarding the other attempts to insert the same transgene in other positions? Also, the reference to the X-shredder here does not seem as relevant as for other approaches such as sex-linked genome editors, e.g. targeting female fertility genes (<https://doi.org/10.1098/rspb.2018.0776>).

3. "Our work lays the groundwork for building X-shredder gene drives in *Drosophila*. We overcome the difficulty of inserting a transgene on the Y by using CRISPR-Cas9 and find a suitable region on the Y to robustly express transgenes."

AND

"The approach to develop the SGyA line here could be used to develop X-shredders or split gene drives for the control of pests with XY sex systems (e.g. *D. suzukii*)."

I have nothing against the statements however it should be made clearer that the main cornerstone, not yet addressed in this or other published works, is the ability to circumvent MSCI acting against meiotic specific promoters (necessary to successfully engineer the X-shredder gene drive) once these are inserted in the Y (or X) chromosome. Also, the promoter used here is premeiotic and therefore expected to be unsuitable for meiotic X-shredding to generate male biased progenies. I would strongly encourage the authors to clarify this aspect unless able to provide supporting evidence of meiotic expression from the Y locus investigated here.

4. "This revealed that the gRNA/effector allele persists in the population for longer for the Y-linked system, which could be useful for achieving more enduring population protection from vector-borne disease transmission, for instance."

I believe that this part of the modeling, which is also more relevant to the work done here, should be expanded further. For example, would this statement be generally applicable or particularly relevant to specific applications and targets? What is the genetic explanation for the prolonged persistence? Most of these aspects may be known or more or less obvious within the genetic control field but it is unlikely that this will be the case for the broader research community.

5. "Next, we compared the performance of a Y-linked X-shredder to an autosomal homing-based drive targeting a gene required for ♀ fertility 8. This revealed that the Y-linked X-shredder achieves population elimination more quickly than the autosomal homing-based design. While resistant alleles can lead to a population rebound for both systems, Y-linked X-shredders can be engineered to target multiple loci on the X chromosome simultaneously, providing a clear route to minimizing resistant allele generation. Thus, for both suppression and modification, there are clear use cases and benefits to inserting drive components on the Y chromosome."

Not convinced that this is adding anything novel/unknown, at least as it is written now, or particularly relevant to this work at this stage, yet relatively far from being an active Y-linked X-shredder (see comment for point 3).

Minor comments typos

- I-PPoI should be I-PpoI

- GSS and LBM abbreviations need full names when first used.

REVIEWERS' COMMENTS

Reviewer #1 (Remarks to the Author):

I am pleased that the authors removed the modeling of *Aedes aegypti* and replaced it with a species that contains a Y chromosome and their choice of *Anopheles gambiae* is perfect.

Thank you for this suggestion.

I am less pleased with their response about using only a single mutant upon which to base their analysis. I completely understand the profound difficulty of engineering the Y chromosome and I applaud their ability to achieve this. Logically speaking, they should be able to do so again, and perhaps a third time, but I do understand this will take time. From my perspective it remains the desired course. In their rebuttal the authors point out that it will at least be equally difficult to achieve similar modifications of the Y chromosome of pest insects but clearly this is the direction they wish to go and it is the rationale for the research. So I would consider this question from the perspective of readers who wish to follow their path. They would be best served by multiple mutants. If this cannot be achieved here then the least the authors can do is to include the description of their "failed" mutants (that is, those described in their rebuttal) in the paper itself so that these readers can get an estimate of the pitfalls ahead. I think this also serves the publisher well since it makes the paper more informative and useful from this practical sense.

Thank you for this suggestion - we have now included a description of additional attempts to insert Cas9 on the Y into the manuscript as suggested.

Reviewer #2 (Remarks to the Author):

This is a scientifically sound and useful work and I believe that the manuscript will be of interest in the field as well as suitable for the Nature Communications audience. The current version seems reasonably improved, particularly in terms of clarity. There are only a few aspects that I would recommend the authors to clarify in the discussion and/or relevant sections of the manuscript before publication.

Thank you for this feedback - see below for details on how we addressed each point.

1. "In addition to this local effect on gene expression, a global effect of the Y chromosome termed meiotic sex chromosome inactivation 62, may also be playing a role. These phenomena are likely involved in our observations regarding reduced marker gene expression in some SGyA ♂'s (Fig. S1)."

I believe that the authors here may want to refer to the sex chromosome-wide transcriptional suppression rather than meiotic silencing as the latter would have no effect (at least not on its own) on the expression of the 3xP3 driven RFP marker reported in S1 but only on the Cas9/GFP expressed in the germline.

Thank you for this suggestion - we have incorporated the possibility of sex chromosome-wide transcriptional suppression playing a role in silencing as well.

2. "As such, optimization may require inserting our transgene in another location on the Y or using alternative insulators 63. In either case, our SGyA line enables the male-only transmission of the Cas9 component in a split drive

system which allows for expression and gene drive conversion only in males thereby preventing the accumulation of resistant alleles that result from maternal transmission 8,18,26,36. This feature is critical for male-gene drive systems like X-shredders.”

There is little discussion or reference in previous sections about the utility of insulators as well as regarding the feasibility of targeting other Y loci. Probably useful to include in the supplementary some more information regarding the other attempts to insert the same transgene in other positions?

Thank you for this feedback. We have now added data on other Y-insertion sites we attempted. We also added in a bit more information regarding the utility of insulators:

“We also incorporated a tdTomato transformation marker driven by the eye-specific 3xP3 promoter and flanked by *gypsy* and CTCF insulators to improve overall expression levels by acting as barrier elements that can block the propagation of heterochromatic structures into adjacent euchromatin ²⁹.”

Also, the reference to the X-shredder here does not seem as relevant as for other approaches such as sex-linked genome editors, e.g. targeting female fertility genes (<https://doi.org/10.1098/rspb.2018.0776>).

Thank you for this feedback - we have added in sex-linked genome editors as another example and cited the paper as suggested.

3. “Our work lays the groundwork for building X-shredder gene drives in *Drosophila*. We overcome the difficulty of inserting a transgene on the Y by using CRISPR-Cas9 and find a suitable region on the Y to robustly express transgenes.” AND “The approach to develop the SGyA line here could be used to develop X-shredders or split gene drives for the control of pests with XY sex systems (e.g. *D. sukukii*).”

I have nothing against the statements however it should be made clearer that the main cornerstone, not yet addressed in this or other published works, is the ability to circumvent MSCI acting against meiotic specific promoters (necessary to successfully engineer the X-shredder gene drive) once these are inserted in the Y (or X) chromosome. Also, the promoter used here is premeiotic and therefore expected to be unsuitable for meiotic X-shredding to generate male biased progenies. I would strongly encourage the authors to clarify this aspect unless able to provide supporting evidence of meiotic expression from the Y locus investigated here.

Thank you for this suggestion. We have added these points to the discussion:

“It is well known that in *Drosophila melanogaster* the position of a gene within euchromatic/heterochromatic regions can contribute to regulating expression. This phenomenon, called Position Effect Variegation (PEV)⁶², may explain the patchy expression of the fluorescent marker often observed in the eyes, and the overall reduced expression of Cas9 protein from the SGyA ♂’s as compared to an autosomal source of Cas9. In addition to this local effect on gene expression, a global effects on the Y chromosome including meiotic sex chromosome inactivation (MSCI)⁶³ and sex chromosome-wide transcriptional suppression may also be contributing to our observations regarding reduced gene expression in some SGyA ♂’s (Fig. S1). Despite reduced gene expression in some individuals, our SGyA line still generated robust Cas9 expression in the germline that was capable of generating heritable mutations (Fig. S1C). ”

AND

“It should be noted that the Vasa promoter is generally premeiotic and therefore will not be suitable for meiotic X-shredding to generate male biased progenies. That said, future efforts to circumvent MSCI acting against meiotic specific promoters should be undertaken as this will be a necessary cornerstone to successfully engineer the X-shredder gene drive once these are inserted in the Y (or X) chromosome”

4. “This revealed that the gRNA/effector allele persists in the population for longer for the Y-linked system, which could be useful for achieving more enduring population protection from vector-borne disease transmission, for instance.”

I believe that this part of the modeling, which is also more relevant to the work done here, should be expanded further. For example, would this statement be generally applicable or particularly relevant to specific applications and targets? What is the genetic explanation for the prolonged persistence? Most of these aspects may be known or more or less obvious within the genetic control field but it is unlikely that this will be the case for the broader research community.

We agree with the reviewer that expanding on this part of the modeling would be of interest, but consider it to be beyond the scope of this paper and something that we plan to investigate further in a subsequent modeling-focused paper. We have added the text *“for default parameters”* to highlight that, as the reviewer suggests, the general applicability of this result will need to be explored further. In response to the reviewer’s other suggestion, a genetic explanation is provided for the prolonged persistence: *“the gRNA/effector allele persists in the population for longer for the Y-linked system, partly due to the fact that female fertility costs are not manifest for Y-linked Cas9 alleles.”*

5. “Next, we compared the performance of a Y-linked X-shredder to an autosomal homing-based drive targeting a gene required for ♀ fertility 8. This revealed that the Y-linked X-shredder achieves population elimination more quickly than the autosomal homing-based design. While resistant alleles can lead to a population rebound for both systems, Y-linked X-shredders can be engineered to target multiple loci on the X chromosome simultaneously, providing a clear route to minimizing resistant allele generation. Thus, for both suppression and modification, there are clear use cases and benefits to inserting drive components on the Y chromosome.”

Not convinced that this is adding anything novel/unknown, at least as it is written now, or particularly relevant to this work at this stage, yet relatively far from being an active Y-linked X-shredder (see comment for point 3).

Thank you for your feedback. We would prefer to leave this result in the manuscript as not all readers will know the performance differences between a Y-linked X-shredder and an autosomal homing-based design. Keeping these results in this figure enables the reader to readily draw comparisons between these systems, illustrating the comparative utility of Y-linked systems.

Minor comments typos

- I-PPol should be I-Ppol

Corrected

- GSS and LBM abbreviations need full names when first used.

Corrected